

# Interaction of microphysics and dynamics in a warm conveyor belt simulated with the ICON model

Annika Oertel[1], Annette K Miltenberger[2], Christian M Grams[1], and Corinna Hoose[1]

[1]Institute of Meteorology and Climate Research, Karlsruhe Institute of Technology
[2]Institute for Atmospheric Physics, Johannes Gutenberg University Mainz

**Correspondence:** Annika Oertel (annika.oertel@kit.edu)

**Abstract.** The representation of warm conveyor belts (WCBs) in numerical weather prediction (NWP) models is important, as they are responsible for the major precipitation in extratropical cyclones and modulate the large-scale flow evolution. Their cross-isentropic ascent into the upper troposphere is influenced by latent heat release mostly, but not exclusively, from cloud formation whose representation in NWP models is associated with large uncertainties. The diabatic heating additionally mod-

ifies the potential vorticity (PV) distribution which influences the circulation. We analyse diabatic heating and associated PV rates from all physics processes, including microphysics, turbulence, convection, and radiation, in a case study of a WCB that occurred during the North Atlantic Waveguide and Downstream Impact Experiment (NAWDEX) campaign using the Icosahedral Nonhydrostatic (ICON) modelling framework. In particular, we consider all individual microphysical process rates that are implemented in ICON's two-moment microphysics scheme, which sheds light on (i) which microphysical processes dominate

the diabatic heating and PV structure in the WCB, and thus, potentially influence the large-scale flow, and (ii) which microphysical processes are most active during the ascent and influence cloud formation and characteristics. For this purpose, diabatic heating and PV rates are integrated along online WCB trajectories. Our convection-permitting simulation setup also takes the reduced aerosol concentrations over the North Atlantic into account. Complementary Lagrangian and Eulerian perspectives on diabatic heating and PV modification confirm that microphysical processes are the dominant diabatic heating contribution

during ascent. Near cloud top longwave radiation cools WCB air parcels. Due to the longevity of the WCB cloud band, the diabatic heating contributions from radiation, and corresponding PV modification in the upper troposphere, are non-negligible. The turbulence scheme is active in the WCB ascent region, despite large gradient Richardson numbers, and process rates from turbulence and microphysics partially counteract each other. From all microphysical processes condensational growth of cloud droplets and vapor deposition on frozen hydrometeors most strongly influence diabatic heating and PV, while below-cloud

evaporation strongly cools WCB air parcels prior to their ascent and increases their PV value. PV production is strongest near surface, and extends up to 4 km height with substantial contributions from condensation, melting, evaporation, and vapor deposition. In the upper troposphere, PV is reduced by diabatic heating from vapor deposition, condensation, and radiation. Activation of cloud droplets as well as homogeneous and heterogeneous freezing processes have a negligible diabatic heating contribution due to small overall mass conversion, but their detailed representation is likely important as the hydrometeor

size distributions influence other microphysical processes. Generally, faster ascending WCB trajectories are heated markedly more than more slowly ascending WCB trajectories, which is linked to larger initial specific humidity content of fast WCB





trajectories providing a thermodynamic constraint on total microphysical heating. Yet, the total diabatic heating contribution of convectively ascending trajectories is relatively small due to their small fraction in this case study.

# 1 Introduction

Extratropical cyclones (ETCs) are often characterized by a large-scale and vertically-extended cloud band with a long-lived cirrus cloud cover, which is formed by a strongly ascending airstream, the so-called warm conveyor belt (Wernli and Davies, 1997; Eckhardt et al., 2004; Madonna et al., 2014; Oertel et al., 2019). WCBs are responsible for the major part of precipitation associated with ETCs (Pfahl et al., 2014), they are dynamically important airstreams as they can interact with and modify the upper-level flow (Grams et al., 2011; Rodwell et al., 2017; Grams et al., 2018; Berman and Torn, 2019; Oertel et al., 2023),

and they largely influence the cloud radiative forcing in the mid-latitudes (Joos, 2019). During the WCB's ascent from the lower into the upper troposphere a wide range of cloud processes take place, ranging from warm-phase to mixed-phase and ice-phase processes (Forbes and Clark, 2003; Joos and Wernli, 2012; Crezee et al., 2017; Gehring et al., 2020; Mazoyer et al., 2021). The representation of these microphysical processes in numerical weather prediction (NPW) models, in particular the parameterisation of ice-phase and mixed-phase processes, is still highly uncertain (e.g. Khain et al., 2015; Dearden et al., 2016;

Hande and Hoose, 2017; Field et al., 2017).

The model representation of cloud processes influences cloud characteristics, dynamics and surface precipitation, and hence, model uncertainty may modify the WCB's cloud and precipitation structure. For example, cloud condensation nuclei (CCN) concentrations affect cloud internal dynamics and surface precipitation, which has been studied extensively for convective systems (e.g. Seifert and Beheng, 2006b; Noppel et al., 2010; Barthlott and Hoose, 2018; Igel and van den Heever, 2021;

Barthlott et al., 2022). The response to changing CCN concentrations is not unidirectional and depends on the applied NWP model and microphysics scheme, as well as on the synoptic conditions (Tao et al., 2012; Fan et al., 2016). Due to the very different dynamics and associated microphysical process pathways between convective systems and large-scale WCB ascent in extratropical cyclones, these results cannot directly be transferred to WCBs where updraft velocities are substantially smaller and markedly influenced by synoptic conditions. A climatological comparison of precipitation characteristics during WCB

ascent in polluted versus clean conditions in a simulation with a relatively low spatial resolution of $1.875° \times 1.875°$ showed only a small impact of aerosols on WCB associated surface precipitation (Joos et al., 2017), whereby polluted conditions slightly delayed precipitation formation. Observations suggest that the cloud liquid water path in WCBs is influenced to some extent by the cloud droplet number concentration (CDNC), which modifies cloud albedo (McCoy et al., 2018), and thus the cloud radiative forcing. This relation could also be represented in idealized numerical model simulations with varying prescribed

aerosol concentrations (McCoy et al., 2018). Generally, cloud radiative forcing is substantially influenced by microphysical and macrophysical properties of cirrus clouds in the upper troposphere (e.g., Zhang et al., 1999; Joos et al., 2014), which to some extent depends on the formation pathways of the cirrus clouds (Krämer et al., 2016; Luebke et al., 2016). The presence of different ice formation pathways in WCB-related ice clouds (Wernli et al., 2016) emphasizes the relevance of the detailed representation of ice nucleation in WCBs. Assumptions about ambient aerosol concentrations and characteristics are one source





of parametric uncertainty in most current NWP systems (or of initial condition uncertainty in aerosol-coupled systems), but the representation of many other microphysical quantities and processes is also subject to parametric (and systematic) uncertainty, and thereby, contributes to the overall uncertainty in NWP.

Aside from modifying cloud properties, microphysical processes and their associated latent heating play an important role for dynamics of extratropical cyclones (e.g., Reed et al., 1992; Rossa et al., 2000; Wernli et al., 2002; Binder et al., 2016). Uncertainty in the representation of latent heat released during cloud formation can represent a dominant source of initial error growth (Baumgart et al., 2019). In general, WCBs influence the large-scale flow through both direct PV modification in the lower and upper troposphere (e.g., Rossa et al., 2000; Pomroy and Thorpe, 2000; Joos and Forbes, 2016; Oertel et al., 2020; Mazoyer et al., 2021) and their related divergent outflow in the upper troposphere (e.g., Grams and Archambault, 2016; Steinfeld and Pfahl, 2019; Teubler and Riemer, 2021).

The WCB ascent strength and its isentropic outflow level, which are tightly related to latent heat release, strongly impact the upper-level divergent wind field. It has been shown that uncertainty in the representation of latent heating substantially influences the WCB's cross-isentropic ascent strength (e.g., Martínez-Alvarado et al., 2014; Joos and Forbes, 2016; Mazoyer et al., 2021) and the frequency of WCB ascent (Pickl et al., 2022). A misrepresentation of WCB ascent, and its isentropic outflow level, can influence the jet stream wind speed (Schäfler and Harnisch, 2015) and also lead to forecast busts through underestimation of the downstream ridge amplitude (Grams et al., 2018).

In addition to the impact on ascent strength, diabatic heating modifies the PV distribution in extratropical cyclones (see, e.g., Wernli and Davies, 1997; Joos and Wernli, 2012; Spreitzer et al., 2019; Attinger et al., 2019; Rivière et al., 2021, for more details). On larger scales, where vertical diabatic heating gradients dominate, diabatic heating during slantwise WCB ascent increases low-level PV values, which can lead to an intensification of cyclones (Rossa et al., 2000; Joos and Wernli, 2012; Binder et al., 2016). In the upper troposphere, PV values are decreased, which can modify the upper-level flow evolution (e.g., Wernli and Davies, 1997; Pomroy and Thorpe, 2000; Grams et al., 2011; Madonna et al., 2014; Methven, 2015; Saffin et al., 2021). Hence, as ascending air parcels pass through the quasi-vertical PV dipole related to mid-tropospheric diabatic heating from cloud formation, their PV values on average increase before they decrease to values close to their initial PV value (e.g., Madonna et al., 2014; Methven, 2015). On smaller-scales, horizontal diabatic heating gradients become increasingly important and form quasi-horizontal PV dipoles in an environment where vertical wind shear is aligned with smaller-scale horizontal diabatic heating gradients (Chagnon and Gray, 2009; Harvey et al., 2020; Oertel et al., 2020, 2021). The latter is particularly relevant for convective activity embedded in the WCB airstream (Oertel et al., 2020, 2021). Case studies (Joos and Wernli, 2012; Joos and Forbes, 2016; Mazoyer et al., 2021) showed that the major contribution to the PV modification results from condensation and vapor deposition on snow and ice in the lower and upper troposphere, respectively. The underestimation of upper-level PV reduction in an autumn WCB case study was linked to an underestimation of vapor deposition on ice during WCB ascent (Mazoyer et al., 2021), and points towards the importance of the model representation of ice-phase processes.

In summary, uncertainty related to the parameterisation of microphysical processes influences the WCB cloud structure as well as the associated diabatic heating and PV structure, both important factors for NWP. Previous case studies applying one-moment or quasi two-moment microphysics schemes show that the detailed WCB ascent and net diabatic heating vary





depending on the applied microphysics scheme, the NWP model, and the WCB case (Martínez-Alvarado et al., 2014; Joos and Forbes, 2016; Mazoyer et al., 2021; Choudhary and Voigt, 2022). An improved understanding of individual processes relevant for the cloud structure and diabatic heating in a WCB, will facilitate the identification of microphysical processes and related model parameters that are uncertain but particularly relevant for WCB ascent.

Building upon previous studies, we provide detailed insight in relevant microphysical process rates in the ICON model
(Zängl et al., 2015), their interactions, and partially compensating processes in a two-moment microphysics scheme, which may - aside from enhanced physical understanding - ultimately contribute to model development and ensemble configuration (e.g., Ollinaho et al., 2017). Specifically, we combine Lagrangian and Eulerian perspectives to quantify diabatic heating and PV rates considering diabatic heating contributions from all physics parameterisations with a strong focus on individual microphysical process rates represented in ICON's two-moment scheme (Seifert and Beheng, 2006a).

In this study, we address the following questions:

1. Which non-conservative processes dominate the total diabatic heating in this WCB case study?

2. Which microphysical processes are active during WCB ascent in ICON's two-moment scheme, and how large are their individual contributions to the total microphysical diabatic heating?

3. What is the influence of the non-conservative processes on the PV distribution in the WCB?

The study is structured as follows: We first introduce the ICON model before introducing the applied Lagrangian and Eulerian diagnostics (Section 2). Section 2 also shows the modifications to the CCN activation scheme to account for reduced aerosol concentrations in the North Atlantic region. Next, we introduce the WCB case study (Section 3) and quantify the total integrated diabatic heating associated with the individual (micro-)physics processes (Section 4). Before concluding with a summary (Section 6), we show the processes that impact the spatio-temporal PV structure of the WCB using complementary
Lagrangian and Eulerian perspectives (Section 5).

## 2   Methods

Diabatic processes in a WCB case study and their impact on the PV structure are investigated in a high-resolution ICON simulation. Lagrangian metrics providing insight into WCB air parcel history are combined with an Eulerian perspective providing spatial context for the WCB air parcel analysis. The following describes the numerical model including the individual
microphysical processes from the two-moment scheme (Sections 2.1-2.2) and model diagnostics used in our analysis, i.e. diabatic heating and PV rates (Section 2.4), and online trajectories (Section 2.5). Details on the composite analysis of WCB related cloud characteristics used for the Eulerian analysis are provided in Section 2.6. To account for aerosol concentrations over the North Atlantic we modify the CCN activation scheme, which is described in Section 2.3.



## 2.1 ICON model setup

We employ the Icosahedral Nonhydrostatic (ICON) modelling framework (version 2.6.2.2; Zängl et al., 2015) to simulate a WCB case that occurred in the North Atlantic in October 2016 during the NAWDEX campaign (IOP 7; Schäfler et al., 2018). We run a global simulation with a R03B07 grid (approx. 13 km effective grid spacing, corresponding to the operational resolution of the global ICON model setup used by the German weather service) and with a time step of 120 s. Additionally, two higher resolution refined domains are nested in the global simulation on R03B08 and R03B09 grids (approx. 6.5 km and 3.3 km effective grid spacing, Fig. 1). The time step for the refined nests is by definition halved from the respective lower resolution parent domain, i.e., 60 s and 30 s. The higher resolution simulations are coupled by two-way nesting to the next lower resolution simulation, i.e., the lower resolution simulation provides lateral boundary conditions for the next higher resolution nest at every time step, and in turn, the lower resolution prognostic fields are nudged towards the solution from the respective higher resolution. The two refined nests were each chosen to encompass (i) the major part of WCB inflow, ascent and outflow region, and (ii) the WCB ascent region (Fig. 1). The vertical dimension is represented by terrain-following smooth level vertical (SLEVE) coordinates (Leuenberger et al., 2010) with 90 levels. Level number is identical in all nests, but the coordinate systems in higher-resolution nests reflect the higher resolution of topography. The simulation is initialized from the ECMWF analysis at 18 UTC 03 October 2016 and runs freely for 72 h. Deep convection is treated explicitly in the nested domains, and the Tiedtke-Bechtold convection scheme is used for the global domain (Tiedtke, 1989; Bechtold et al., 2008; ECMWF, 2016). Shallow convection is parameterised in all domains. Turbulence (Raschendorfer, 2018) and sub-grid scale orographic drag (Lott and Miller, 1997) and non-orographic gravity wave drag (Orr et al., 2010) are parameterised by the standard schemes used in ICON. Radiation is described by the Rapid Radiative Transfer Model (RRTM; Mlawer et al., 1997). Cloud microphysical processes are represented by a two-moment microphysics scheme (Seifert and Beheng, 2006a) with six hydrometeor types (cloud, ice, rain, snow, graupel, and hail) and prognostic variables for the hydrometeor mass mixing ratios (cloud liquid ($q_c$), ice ($q_i$), rain ($q_r$), snow ($q_s$), graupel ($q_g$), and hail ($q_h$)), and corresponding number concentrations. A detailed description of the microphysical processes analysed in-depth in our study is provided in the next section.

## 2.2 Microphysical processes

We focus on the microphysical processes that are associated with phase changes of water, and hence diabatic heating or cooling (and do not explicitly analyse adiabatic processes such as accretion, autoconversion, or ice collision processes that are not related to latent heat release). In the following, the microphysical processes are listed in the order in which they are called in ICON. We additionally consider temperature tendencies from the other physics parameterisations in ICON, which are also included in the list below.

– Saturation adjustment I (SATAD I)

The saturation adjustment removes sub- or super-saturation with respect to liquid water and establishes thermodynamic equilibrium between water vapor and liquid water by condensation or evaporation of cloud droplets, respec-





tively, assuming that clouds relax instantly to a thermodynamic equilibrium. In ICON the saturation adjustment is called twice: Once before the explicit microphysical processes, and once afterwards.

– Turbulence scheme (TURB)

Turbulent diffusion in the free troposphere is represented by a $2^{nd}$ order closure scheme developed by Raschendorfer (2018) which uses a prognostic equation for turbulent kinetic energy.

– CCN activation (QCACT)

The activation of CCN is based on a modification of the Hande et al. (2016) parameterisation and described in detail below (see Section 2.3 for details).

– Homogeneous and heterogeneous ice nucleation (QIFRZ)

Homogeneous freezing is parameterised following Kärcher and Lohmann (2002) and Kärcher et al. (2006). The heterogeneous freezing parameterisation is adapted from Hande et al. (2015) and includes parameterisations for immersion and deposition freezing. Their freezing rates increase with decreasing temperature. If cloud droplets are present, immersion freezing is active below -12°C. Deposition nucleation takes place in a temperature range from -20°C to -53°C and is scaled by grid-scale variables, such as supersaturation w.r.t. ice.

– Homogeneous freezing of cloud droplets (QCFRZ)

Freezing of cloud droplets is described by a homogeneous freezing rate increasing exponentially with decreasing temperature (Jeffery and Austin, 1997). Below -50°C all cloud droplets freeze immediately.

– Depositional growth of all ice-phase hydrometeors (QXDEP)

Deposition of vapor on and sublimation from frozen hydrometeors is computed with a relaxation time-scale approach (Morrison et al., 2005). The deposition parameters are calculated for the four ice hydrometeor types (ice, snow, graupel, and hail) explicitly. We individually consider diabatic heating from vapor deposition on ice (QIDEP), snow (QSDEP), and the sum of vapor deposition on graupel and hail (QGDEP).

– Riming (QXRIM)

Riming is treated explicitly between the individual hydrometeor types (see Seifert and Beheng, 2006a). A fraction of the rimed mass is redistributed to smaller ice particles if the temperature conditions for the Hallet and Mossop ice multiplication process are met, but this is not associated with any additional latent heat release. We consider heating from the sum of all individual riming processes.

– Freezing of rain (QRFRZ)

The hetereogeneous freezing rate of rain increases with decreasing temperature and increasing mean rain drop mass following Bigg (1953). Below -40°C all rain instantly freezes.

– Melting of ice hydrometeors (QXMLT)

Above 0°C frozen hydrometeors melt. Melting rates for snow, graupel and hail are represented explicitly, while all cloud ice melts immediately. Above 10°C all snow immediately melts, too. We consider the sum of all melting processes.



– Evaporation of melting ice hydrometeors (QSGHEVAP)

Above 0°C, liquid surfaces of melting ice hydrometeors can evaporate (Seifert and Beheng, 2006a). Evaporation of melting ice hydrometeors is treated similar to evaporation of rain with a constant surface temperature of 273 K. Again we consider the sum of the processes rates for snow, graupel, and hail.

– Evaporation of rain (QREVAP)

Evaporation of rain in subsaturated conditions is based on a parameterisation by Seifert (2008) taking into account the rain drop size distribution which, e.g., allows smaller rain drops to evaporate faster.

– Second saturation adjustment (SATAD II)

The second saturation adjustment after the explicit microphysical processes is called to start the subsequent slow-physics parameterisations in an equilibrated state.

– Slow physics parameterisations

The slow physics processes are only called at a reduced frequency, and include parameterisations of convection (CON, dt=600 s), radiation (RAD, dt=1800 s, and non-orographic and orographic gravity wave drag (DRAG, dt=600 s). Furthermore, radiation is calculated from a coarser grid (R03B06). Diabatic temperature tendencies from these processes are also considered in our analysis.

## 205   2.3   CCN activation

The recently developed CCN activation scheme (Hande et al., 2016) for the ICON-DE domain is optimized for aerosol number concentrations over Germany and based on extensive measurements of aerosol size distributions during the HD(CP)[2] Observational Prototype Experiment (HOPE) field campaign. The CCN activation is parameterised as function of updraft velocity and pressure, accounting for thermodynamic conditions and vertical distributions of aerosols, respectively. Generally, CCN

concentration is high in the boundary layer and strongly decreases above.

Aerosol number concentrations over continental regions can differ substantially from concentrations over the open ocean. Previous studies showed that aerosol number concentrations over ocean decrease less strongly with height than over continental regions, especially in the lower troposphere (Hudson and Xie, 1999; Wang et al., 2021), although variability in aerosol amount and its vertical distribution is large and, among others, varies with season and synoptic conditions. In WCBs the cloud liq-

uid path, cloud cover and albedo can be sensitive to the representation of cloud droplet number concentration, which is tightly linked to the representation of aerosol concentrations and CCN activation (McCoy et al., 2018). To account for differing aerosol characteristics over the North Atlantic compared to continental Germany, we modified the CCN activation parameterisation taking into account airborne measurements of size-resolved aerosol number concentrations from three research flights of the British Facility for Airborne Atmospheric Measurements (FAAM) BAe 146 that took place during the NAWDEX campaign

in September and October 2016 (CEDA, Facility for Airborne Atmospheric Measurements; Natural Environment Research Council; Met Office, 2016a, b, c; CEDA, Centre for Environmental Data Analysis, 2016). Size-binned aerosol number con-



centrations were measured with a Passive Cavity Aerosol Spectrometer Probe (PCASP) in the range of 0.2 $\mu$m to 3 $\mu$m (Nott, 2013; Droplet Measurement Technologies, 2017).

FAAM flights started at Prestwick airport (Scotland), and measurements were taken to the West and North of the United
Kingdom over the Atlantic and Scotland (Fig. 2a). To account for air mass origin of PCASP measurements, 24-h backward trajectories were calculated with LAGRANTO (Sprenger and Wernli, 2015) from 3-h ERA5 reanalysis data (Hersbach et al., 2019, 2020). Trajectories were started in the region around the flight tracks and every 100 hPa in the vertical. The backward trajectories show that the flights in September were influenced by long-range transport of maritime air advected from the North Atlantic, while air measured during the October flight is potentially more strongly influenced by continental aerosol
concentrations with low-level and mid-level air masses advected from the North Sea region (Fig. 2a).

PCASP measurements with a temporal resolution of 1 s are available. Data during strong sinking or rising flight segments was removed, which substantially reduced the number of available data (Fig. A1a). 10-minute along-flight averages were calculated prior to computing the median size distribution for the marine boundary layer (MBL) and the free troposphere (FT), separately.

Both aerosol size distributions show two local maxima, hence, two log-normal aerosol size distributions for the accumulation and the coarse mode were fitted (Fig. A1b). Assuming ammonium sulfate for the accumulation mode and sea salt for the coarse mode (note that results are not very sensitive to these assumptions), parcel model simulations with the cloud parcel model *pyrcel* (Rothenberg and Wang, 2016) were performed to obtain activated cloud droplet number concentrations (CDNC). Simulations of rising parcels were started at different pressure values (1000 hPa - 400 hPa) with constant vertical velocities of (0.01 ms$^{-1}$
to 50 ms$^{-1}$). Initial temperature values for the parcel simulations follow a dry adiabatic temperature profile and initial relative humidity is set to 99%. Parcel model simulations are computed for 3000 s and the CDNC values at the time of maximum supersaturation are retrieved. The resulting CDNC values were used to fit a parameterisation for CCN activation depending on pressure and vertical velocity (eq. 1), similar to the CCN activation scheme developed by Hande et al. (2016). CCN number concentrations $N_c$ are assumed to be quasi constant in the well-mixed boundary layer, and decay exponentially above with a
scale height of 150 hPa for the accumulation mode and 400 hPa for the coarse mode for a given updraft velocity (Fig. 2b, eq. 1), which follows the aerosol profile. The following equation describes the number of activated cloud droplets $N_c$ as function of vertical velocity $w$ and pressure $p$:

$$N_c(w,p) = N_a(p) \cdot \left(1 + e^{-B(p) \cdot ln(w) - C(p)}\right)^{-1} \tag{1}$$

with

$$N_a(p) = \begin{cases} (250 + 7)\,cm^{-3}, & if\ p \geq 800 hPa \\ (250 \cdot e^{\frac{p-800\,hPa}{150\,hPa}} + 7 \cdot e^{\frac{p-800\,hPa}{400\,hPa}})\,cm^{-3}, & else \end{cases}$$

and



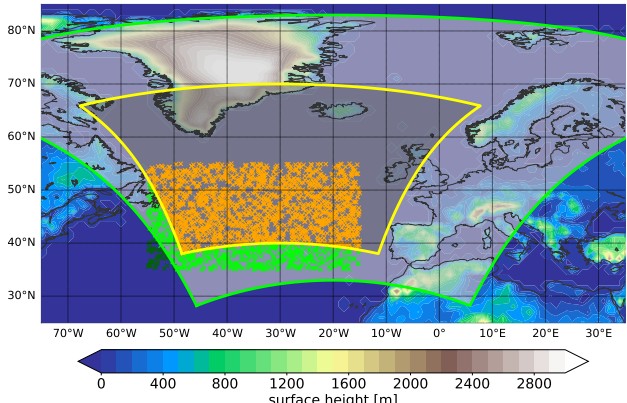

**Figure 1.** Refined nested domains and trajectory starting position. Shown are starting positions in the global domain (R03B07, dark green 'x'), in the first refined nest (R03B08, green 'x'), and in the second refined convection-permitting nest (R03B09, orange 'x'). Domain boundaries of both refined nests are shown in green and yellow. The shading shows the surface altitude (in m).

$$B(p) = b_1 \cdot e^{-b_2 \cdot p + b_3}$$

$$C(p) = c_1 \cdot e^{-c_2 \cdot p + c_3}$$

$$b_1 = 3.46281429e + 00, b_2 = 1.74926665e - 04, b_3 = -2.85967514e - 01$$

$c_1 = 2.72664755e + 00, b_2 = 1.12852352e - 03, c_3 = 1.50069026e + 00.$

CCN activation was limited to temperatures above -38 °C to prevent production of very large ice number concentrations by CCN activation at low temperatures and immediate subsequent homogeneous freezing of the activated cloud droplets. Appendix B provides a comparison of two limited-area simulations applying the modified and original CCN activation schemes.

Although the parameterisation reflects broadly lower CCN concentrations over the North Atlantic compared to Central

Europe (Fig. 2b), CCN concentrations over the North Atlantic are subject to substantial variability (as already evident from the three flights shown in Fig. A1b and used here). This variability cannot be captured by the parameterisation, but for future analysis, we plan to perform systematic sensitivity experiments with varying CCN concentrations taking into account the variability of aerosol number concentrations.

## 2.4 Diabatic heating and PV rates

Latent heat release during phase changes of water is associated with diabatic heating. We calculate potential temperature tendencies from all individual microphysical processes described in Section 2.1 following Joos and Wernli (2012). The diabatic heating rates ($DHR$) are defined for each individual microphysical process as



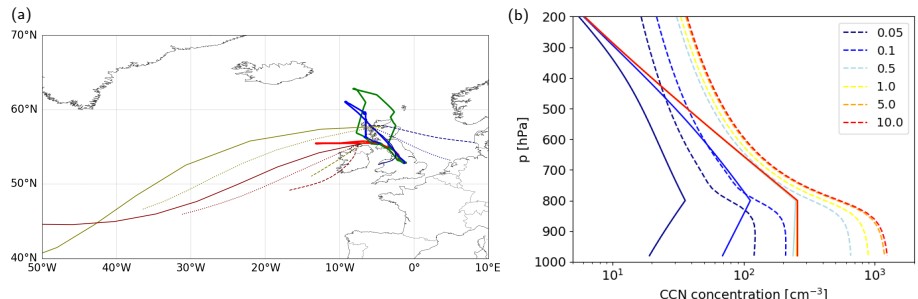

**Figure 2. (a)** Flight tracks for the NAWDEX British FAAM flights on 23 Sep 2016 (red), 27 Sep 2016 (green), and 14 Oct 2016 (blue) including mean 24-h backward trajectories from the flight tracks for all three flights. Trajectory positions are averaged and shown for the lower (dashed lines, $p > 850\,hPa$), middle (dotted lines, $850\,hPa > p > 300\,hPa$) and upper (solid lines, $p < 300\,hPa$) troposphere. **(b)** Vertical profiles of original (dashed) and modified (solid) CCN activation (in $cm^{-3}$) for different vertical velocities (colours, in $m\,s^{-1}$).

$$DHR = \frac{D\theta}{Dt} = S_j * \frac{l_j}{c_p} \cdot \left(\frac{p_0}{p}\right)^\kappa \qquad (2)$$

with the individual microphysical process rates $S_j$ (i.e., the hydrometeor source and sink terms of the respective microphysical processes listed in Table 1 and Section 2.1), and the according latent heat of freezing ($l_f = 3.337 \cdot 1e5\,J\,kg^{-1}$), sublimation ($l_s = 2.8345 \cdot 1e6\,J\,kg^{-1}$) or vaporization ($l_v = 2.5008 \cdot 1e6\,J\,kg^{-1}$), the specific heat at constant pressure $c_p = 1004.64\,J\,kg^{-1}\,K^{-1}$, and the potential temperature $\theta$.

Analogously, the instantaneous PV rates ($PVR$) from individual microphysical processes are calculated (e.g., Joos and Wernli, 2012; Crezee et al., 2017; Spreitzer et al., 2019) as

$$PVR = \frac{DPV}{Dt} = \frac{1}{\rho} \cdot \left(\omega \cdot \nabla \frac{D\theta}{Dt}\right), \qquad (3)$$

with

$$PV = \frac{1}{\rho} \cdot (\omega \cdot \nabla\theta), \qquad (4)$$

where $\rho$ is density, and $\omega$ is the 3D vorticity vector.

For comparison, bulk diabatic heating and related PV tendencies from other parameterisation schemes, i.e. long- and short-wave radiation, turbulence, convection, and drag, are also considered (Table 1). Note that eq. 3 neglects the change of PV from momentum tendencies ($\frac{\partial u}{\partial t}, \frac{\partial v}{\partial t}, \frac{\partial w}{\partial t}$), which can arise from friction, diffusion, and turbulence (Spreitzer et al., 2019; Rivière et al., 2021). Thus, the modification of PV through momentum tendencies that are parameterised in the turbulence, convection and drag schemes are not included in this analysis.





## 2.5 Online trajectories

Although WCBs are large-scale airstreams that can readily be identified with offline trajectories, i.e. trajectories calculated after the model simulation based on low time resolution wind field output, recent studies employed high-resolution online trajectories to represent smaller-scale heterogeneities in the WCB cloud band (e.g., Rasp et al., 2016; Oertel et al., 2020, 2021; Blanchard et al., 2020). These trajectories are computed during the model simulation, i.e. solving the trajectory equation with the actual wind fields at each model timestep. A high temporal resolution of the solution is particularly useful if detailed mi-

crophysical processes are considered (Miltenberger et al., 2016, 2020; Mazoyer et al., 2021). The major advantages of using online trajectories for this study are (i) the wind fields which are used to compute the trajectory position are considered at every model time step, and (ii) the microphysical process rates are integrated online at every model time step (eq. 5). The available online trajectory module in ICON (Miltenberger et al., 2020) is based on the COSMO online trajectory module (Miltenberger et al., 2013, 2014) with modifications to accommodate the unstructured ICON grid.

In the two-way nested simulations we use here, it is not sufficient to follow air parcels in only one nest, if the full evolution of the WCB from pre-ascent to outflow is of interest. As the entire WCB airstream covers approximately 1000 km it is not possible to represent the entire flow feature in the high-resolution nest. Therefore, nesting support was implemented, i.e. trajectories are allowed to enter or leave a refined nest, and are passed on across the border of the nest. The trajectories' positions and properties are always calculated from the resolved wind fields in the nest with the highest spatial resolution at a given

geolocation and using the corresponding model time step. This way, trajectories profit from the high-resolution in the refined nest, however, are not limited to this region. As the trajectory positions are calculated from resolved wind fields, convective updrafts parameterised by the deep convection scheme used in the global domain cannot be explicitly represented.

  Online trajectories are started every 1 h during the 72 h simulation. To reduce the number of trajectories, starting positions

are limited to six vertical levels between 200 and 1500 m, and approximately 1900 longitude and latitude positions pre-defined based on offline WCB calculations with LAGRANTO (Sprenger and Wernli, 2015) for this case study (Fig. 1). The starting region between -55°E to -15°N and 35°N to 55°N encompasses all three domains, however, the largest number of trajectories start in the inner-most convection-permitting nest which covers the main WCB ascent region.

  Along online trajectories we compute the integrated diabatic heating $ADH$ and PV change $APV$, similar to the methodology

developed and applied by (Crezee et al., 2017; Spreitzer et al., 2019; Attinger et al., 2019) for offline trajectories. Accumulated heating and PV change is defined by the sum of instantaneous rates along the path of the trajectory at every domain-specific model time step $\Delta t$, i.e., 120 s, 60 s, and 30 s for the global domain and both refined nests, respectively (see Section 2.1).

$$ADH(\mathbf{x}(t),t) = \sum_{i=1}^{n} DHR(\mathbf{x}(t_i),t_i)\Delta t, \tag{5}$$

$$APV(\mathbf{x}(t),t) = \sum_{i=1}^{n} PVR(\mathbf{x}(t_i),t_i)\Delta t, \tag{6}$$



**Table 1.** List of available variables along online trajectories. The variables are grouped in (i) atmospheric state variables (whose values are output instantaneously), (ii) integrated diabatic heating rates ($ADH$), and (iii) integrated diabatic PV rates ($APV$).

| abbreviation | variable |
| --- | --- |
| lon, lat, alt | longitude, latitude, altitude |
| p, $\rho$, t, $q_v$ | pressure, air density, temperature, specific humidity |
| $q_c, q_r, q_i, q_s, q_g$ | cloud, rain, ice, snow, graupel water content |
| PV | potential vorticity |
| $ADH_{SATADI}$, $ADH_{SATADII}$ | $ADH$ from first and second saturation adjustment |
| $ADH_{QSDEP}$, $ADH_{QIDEP}$, $ADH_{QXDEP}$ | $ADH$ from vapor deposition on snow, ice and sum of all frozen hydrometeors |
| $ADH_{QXRIM}$ | $ADH$ from all riming processes |
| $ADH_{QCACT}$ | $ADH$ from CCN activation |
| $ADH_{QREVAP}$, $ADH_{QSGHEVAP}$ | $ADH$ from evaporation of rain and of melting ice hydrometeors |
| $ADH_{QRMLT}$, $ADH_{QCMLT}$, $ADH_{QXMLT}$ | $ADH$ from melting of frozen hydrometeors into rain and cloud droplets, and the respective sum |
| $ADH_{QRFRZ}$ | $ADH$ from freezing of rain |
| $ADH_{QCFRZ}$ | $ADH$ from homogeneous freezing of cloud droplets |
| $ADH_{QIFRZ}$ | $ADH$ from homogeneous and heterogeneous ice nucleation |
| $ADH_{LW}$, $ADH_{SW}$ | $ADH$ from long- and shortwave radiation |
| $ADH_{CON}$, $ADH_{TURB}$, $ADH_{DRAG}$ | $ADH$ from convection, turbulence, and drag parameterisation |
| $APV_{SATADI}$, $APV_{SATADII}$ | $APV$ from first and second saturation adjustment |
| $APV_{QXDEP}$ | $APV$ from vapor deposition on ice, snow, graupel and hail |
| $APV_{QXRIM}$ | $APV$ from all riming processes |
| $APV_{QCACT}$ | $APV$ from CCN activation |
| $APV_{QREVAP}$ | $APV$ from evaporation of rain |
| $APV_{QRMLT}$ | $APV$ from melting of frozen hydrometeors into rain |
| $APV_{RAD}$, $APV_{CON}$, $APV_{TURB}$ | $APV$ from long- plus shortwave radiation, convection and turbulence parameterisation |

with $i = 1$ corresponding to initialization of each trajectory and $n = t_{end} - t_{init}$ being the length of the trajectory (difference between initialization time $t_{init}$ of each trajectory and end time of the ICON simulation $t_{end}$). Trajectory position and properties (Table 1) are subsequently output every 15 minutes.

From all online trajectories, we select WCB trajectories as trajectories with an ascent rate of at least 600 hPa in 48 h (Wernli
and Davies, 1997; Madonna et al., 2014; Oertel et al., 2019), and subsequently define the ascent phase as the period encompassing the fastest 600 hPa ascent. This 600 hPa ascent phase can be described by a timescale ($\tau_{600}$), which defines the minimum time required to ascend 600 hPa (Oertel et al., 2021).





The start of the fastest 600 hPa ascent ($t_0$) along each trajectory is defined as the start of WCB ascent. To compare the heating among trajectories with differing length prior to $t_0$, $ADH$ is set to 0 at the start of the fastest 600 hPa ascent phase.

## 2.6 Front-relative WCB composites

In addition to the trajectory perspective, we consider cold front-relative vertical cross-section composites across the WCB cloud band, which encompasses a primary WCB ascent region (see Section 3 for details). Figure 3a-c (black markers) illustrates WCB air parcel locations ascending in the cyclones warm sector ahead of the front, and Figure 3d-f shows the corresponding large-scale cloud band. Therefore, West-East oriented vertical cross-section composites are considered, which are centred at the western border of the WCB ascent region identified from Eulerian model fields by the convolutional neural network (CNN) model for WCB ascent 'ELIAS 2.0' (Quinting and Grams, 2022). The CNN model computes conditional WCB ascent probabilities based on the distribution of horizontal wind speed, temperature, geopotential, and specific humidity on selected pressure levels which are taken from the global ICON simulation and were interpolated to a regular 1.0° x 1.0° grid. As shown by Quinting and Grams (2022), 'ELIAS 2.0' adequately represents the WCB ascent footprint and provides a smooth mask of the WCB ascent region. Quinting et al. (2022) compared WCB ascent footprints based on coarse-grained Eulerian fields from a high-resolution ICON simulation with offline trajectories that were computed from a 0.1° x 0.1° grid and found a good agreement between the location of trajectory ascent and the CNN-based footprint. Section 3 and Fig. 3 illustrate that this also applies to the online trajectories.

The vertical cross-section composites are calculated from individual zonal cross-sections through the WCB ascent region ahead of the cold front (here defined as region where the conditional probability of the CNN exceeds 0.3 Quinting and Grams, 2022) during 04 to 05 Oct using data from the first refined nest (R03B08) with a zonal grid spacing of 0.1°. The cross-section composites illustrate the vertical cloud structure associated with the ascending WCB airstream, and demonstrate the spatial distribution of diabatic heating and PV change.





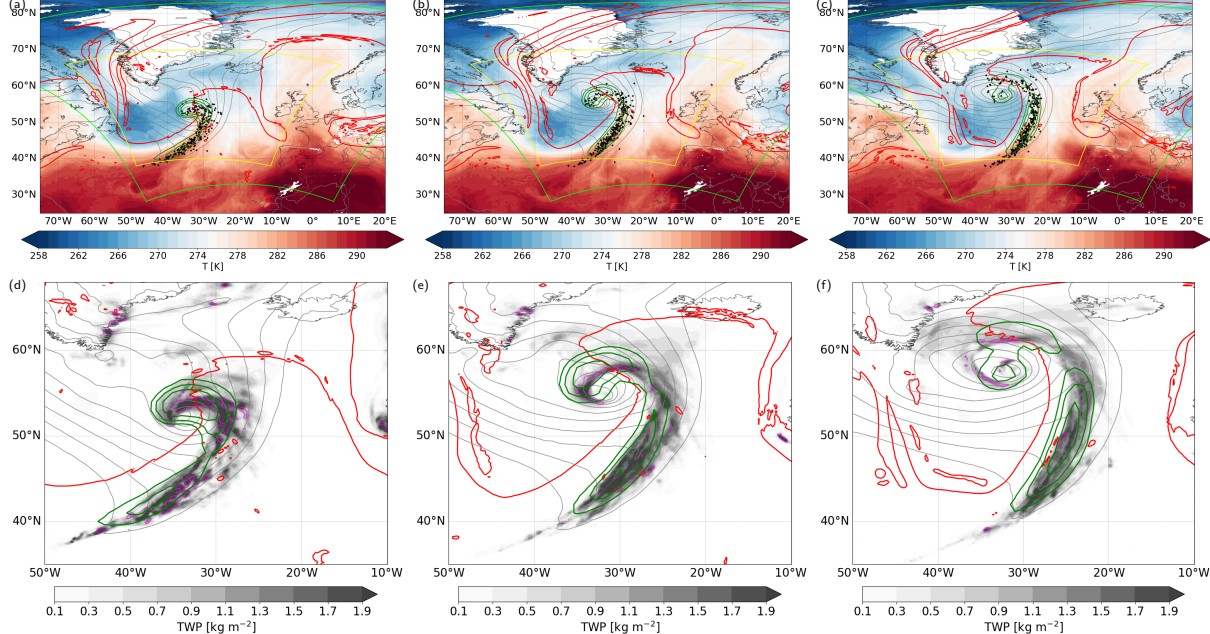

**Figure 3. (a-c)** Temperature at 850 hPa (shading, in K), SLP (grey contours, every 5 hPa), 2 PVU at 320 K isentrope (red contour), WCB ascent probability (at conditional probabilities of 0.3, 0.5, and 0.7, green contours), and online WCB trajectory positions for trajectories ascending at least 25 hPa in 2 h (black markers; only every $50^{th}$ trajectory is shown) for **(a)** 12 UTC 4 Oct, **(b)** 18 UTC 4 Oct, **(c)** 00 UTC 5 Oct. The first and second refined nests are outlined in green and yellow. **(d-f)** Total water path (TWP; shading, in kg m$^{-2}$) and graupel water path (magenta contour, at 0.3 kg m$^{-2}$), SLP (grey contours, every 5 hPa), 2 PVU at 320 K isentrope (red contour), and WCB ascent probability (at conditional probabilities of 0.3, 0.5, and 0.7, green contours) for the same times as shown above.

## 3 Case study introduction

The surface cyclone associated with the WCB initially develops on 02 Oct 2016 as weak SLP depression in a strong baroclinic zone east of -60°E at approximately 45°N and is associated with a local positive low-level PV anomaly. It propagates across the North Atlantic along the baroclinic zone at 45°N and has characteristics of a so-called diabatic Rossby wave (DRW, Boettcher and Wernli, 2013). On 03 Oct, the surface cyclone interacts with the approaching upper-level trough, intensifies further and develops clear warm and cold frontal structures. At this stage, the cyclone forms an extended cloud band which is associated

with poleward directed WCB ascent, and subsequently propagates northward ahead of the upper-level trough. On 04 and 05 Oct, the WCB (represented by green contours in Fig. 3) ascends in the cyclone's warm sector ahead of the upper-level trough and into an amplifying ridge downstream (red contour in Fig. 3) which extends far north to 70°N. The ridge subsequently merged with a precursor ridge built-up by an intense precursor cyclone (Steinfeld et al., 2020; Flack et al., 2021) and contributed to the long-lasting block Thor (Maddison et al., 2019). Here, we focus on the period from 04-06 Oct when the WCB ascent is most

pronounced and forms a large-scale cloud band (Fig. 3d-f).



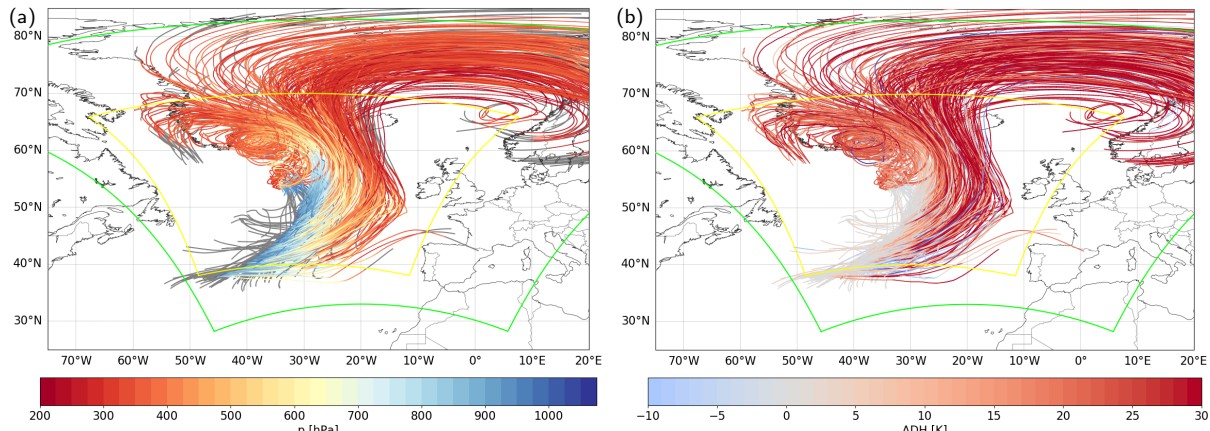

**Figure 4.** Online WCB trajectories starting their fastest 600 hPa ascent on 12 UTC 4 Oct coloured according to **(a)** pressure (p, in hPa) and **(b)** integrated diabatic heating from all microphysical processes along the ascent ($ADH$, in K). In **(a)** the WCB trajectory positions prior to the start of the fastest ascent, and after 48 h ascent time are shown in grey. The refined nests are outlined in green and yellow.

Between 04 and 05 Oct, WCB air parcel locations, identified from online trajectories, ascend poleward in the warm sector of the cyclone ahead of the surface cold front, above the warm front, and near the cyclone center (black markers in Fig. 3a-c). Their ascent is associated with the formation of an extended, dense cloud band (Fig. 3d-f) including the formation of graupel near the frontal regions in the early ascent phase (Fig. 3d). Although WCB trajectories spread out in the ridge once they reached
the upper troposphere, the major ascending motion is confined to the near-frontal regions (Fig. 3a-c), where WCB ascent occurs primarily in a narrow and elongated, north-south oriented band. In general, the location of the ascending segment of the online WCB trajectories is well captured by the CNN-based WCB ascent diagnostic (green contours in Fig. 3a-c), which thus proves useful for diagnosing footprints of WCB ascent from Eulerian data also in high-resolution simulations.

The ascent of the selected approximately 36 000 WCB trajectories takes predominantly place in the region that is covered
by the inner-most convection-resolving nest providing a high resolution in the WCB (Figs. 3a-c and 4). Prior to the start of the WCB ascent, and in particular after the trajectories have reached the upper troposphere near the strong jet region, the number of trajectories that move in or out, respectively, of the inner-most domain increases (Fig. 4). This emphasizes that the transfer of trajectories between different nests is important, in particular if the upstream evolution and the downstream impact of WCB trajectories is of interest. During the transfer of the trajectories across the nest boundaries, no discontinuities in location or
WCB trajectories' properties can be observed, which is illustrated exemplarily for the evolution of pressure (Fig. 4a) and integrated diabatic heating from all microphysical processes along WCB trajectories (Fig. 4b; see Section 4 for a detailed description of diabatic heating along WCB trajectories).

To illustrate the vertical WCB cloud structure and hydrometeor distribution, Figure 5 shows West-East oriented vertical cross-section composites through the WCB cloud band that are centred relative to the western boundary of the CNN based
WCB footprint at relative longitude 0° (Section 2.6). These composites focus on the major WCB ascent region ahead of the



cold front. Figure 5 outlines the vertically-extended cloud band (c) associated with WCB ascent in the warm sector ahead of the sloping cold front (a) and an upper-level trough characterized by a reduced height of the dynamical 2-PVU tropopause (b). The precipitating WCB cloud band, reflected in high values of total hydrometeor content (shading in Fig. 5c) extends from the surface to approximately 11 km height, and slopes towards the East into the upper-level ridge (cf. Fig. 5a,b), which is

characterized by an elevated tropopause height, warmer temperature and low upper-tropospheric PV values (Fig. 5b). Within the WCB cloud band, the 0°C-isotherm is located at approximately 3 km altitude, and the mixed-phase region of the cloud extends from approximately 2 km to 6 km (Fig. 5c). The presence of supercooled liquid water facilitates riming and graupel formation between 2-4 km altitude. The microphysical processes lead to a large net diabatic heating in the cloud, while below the cloud, microphysical processes are associated with a net cooling (Fig. 5d). Diabatic heating contributions from individual

microphysical processes are discussed in Section 4. As expected, the cloud band coincides with ascending motion on the order of 0.01 to 0.08 m s$^{-1}$ averaged across all individual cross-sections, while the post-frontal region west of the cold front is characterized by descent (Fig. 5d). The vertical cross-section composites illustrate the general characteristics of the WCB cloud band and ascent region, however, the compositing technique hides the heterogeneity of WCB ascent behaviour and characteristics reported in previous studies (Oertel et al., 2019; Blanchard et al., 2020; Boettcher et al., in review, 2020; Oertel

et al., 2021). The online trajectories enable a more detailed perspective of the individual processes and detailed WCB ascent behaviour.



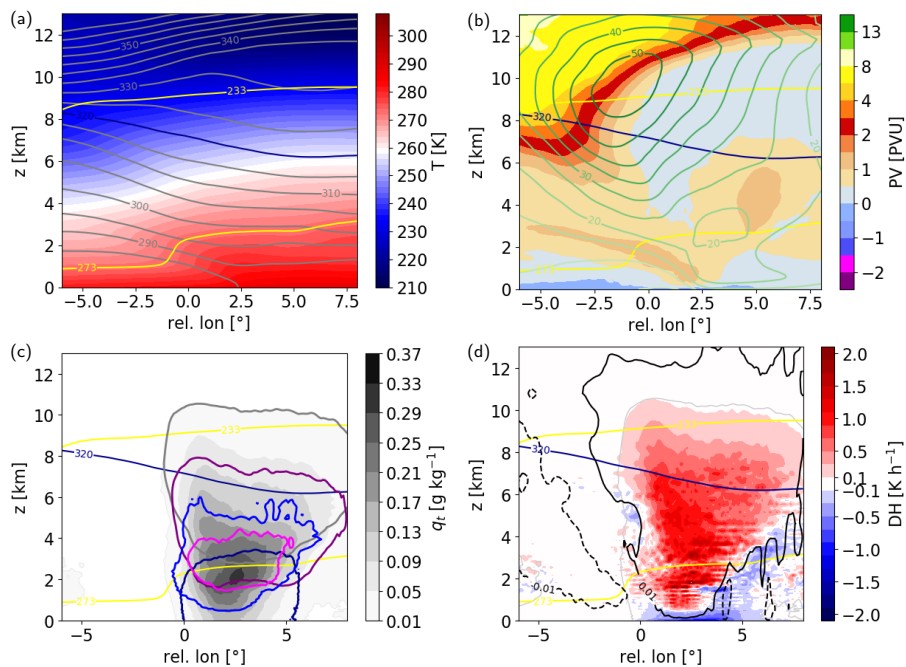

**Figure 5.** Vertical cross-section composites through the WCB ascent region ahead of the cold front for **(a)** temperature (T, in K), and isentropes (every 5 K, grey lines), **(b)** potential vorticity (PV, in PVU) and wind speed (every 5 m s$^{-1}$), **(c)** total hydrometeor content ($q_x$, in g kg$^{-1}$), and rain (blue contour, at 0.02 g kg$^{-1}$), cloud (light blue contour, at 0.02 g kg$^{-1}$), snow (purple contour, at 0.01 g kg$^{-1}$), ice (grey contour, at 0.01 g kg$^{-1}$), and graupel (magenta contour, at 0.02 g kg$^{-1}$) water content, and **(d)** total microphysical diabatic heating (DH, in K h $^{-1}$), vertical velocity (black contours at -0.01 and 0.01 m s$^{-1}$ and outline of WCB cloud (grey line, at $q_x$=0.01 g kg$^{-1}$). All panels show 273 and 233 K isotherms (yellow) and 320 K isentrope (blue).





## 4 Diabatic heating in the WCB

During the trajectories' 48-h ascent from on average 950 hPa to 300-400 hPa (Fig. 6a), they are heated diabatically by on average approximately 25 K (Fig. 6c) resulting in a cross-isentropic ascent from around 290 K to about 315 K (Fig. 6b). However,

substantial variability in ascent behaviour and associated diabatic heating is present. The ascent timescale for 600 hP ascent ($\tau_{600}$) exhibits a large variability, and includes both very rapid convective WCB ascent with timescales of a few hours, and slow and gradual WCB ascent (Fig. 7), which is hidden in the composite perspective. The frequency distribution of $\tau_{600}$ values shows that most WCB trajectories ascend 600 hPa in 20-30 h (Fig. 7a). Thus, the average trajectory ascent time scale is considerable lower than 48 h and amounts to on average 25 h, similar to WCB ascent time scales reported in previous high-resolution case

studies (Rasp et al., 2016; Oertel et al., 2020, 2021; Blanchard et al., 2020). In general, the 600 hPa integrated diabatic heating ranges from 15 K to more than 35 K (Fig. 7a). The latter is only found in rapidly ascending, convective WCB trajectories. Due to the large variability of ascent rates, the entire set of WCB trajectories is stratified into three subsets according to their ascent timescale, which influences the average diabatic heating and detailed microphysical processes. In the following, fast ascent is characterized by $\tau_{600}$ values below 18 h (25th percentile), and slow and gradual ascent is characterized by $\tau_{600}$ values above

30 h (75th percentile). The intermediate subset includes all trajectories with $\tau_{600}$ values in between, and hence, comprises the largest number of trajectories (Fig. 7a). The mean evolution of the three subsets of WCB trajectories is shown in Fig. 6.

As the faster ascending trajectories are heated more strongly, they also reach higher isentropic surfaces (Figs. 7a, and 6b,c) than the slowly ascending trajectories, which emphasizes the large variability within one WCB and is in line with previous case studies (Oertel et al., 2020, 2021). The faster trajectories ascend and also reach the upper troposphere further south ahead

of the upper-level trough.

Recent WCB case studies have highlighted that convective motion can be frequently embedded in the gradual WCB ascent (Rasp et al., 2016; Oertel et al., 2019, 2021; Blanchard et al., 2020, 2021). Yet, the total diabatic heating contribution of embedded convection in WCBs has remained unclear. Figures 7b shows the cumulative distribution of the relative heating contribution of trajectories binned into 1-h $\tau_{600}$ bins. For example, trajectories with a 600 hPa ascent time scale below 10 h

contribute on average 10% to total diabatic heating in this case study (Fig. 7b), although their averaged diabatic heating is substantially larger (Fig. 7c). In contrast, gradually ascending WCB trajectories with ascent time scales above 25 h are on average heated 10 K less but due to their larger number still contribute more than 43% to total heating (Fig. 7b). Selecting thresholds of $\tau_{600}$ of 3 h, 4 h and 7 h (the latter shows a sudden drop of mean $ADH$ in this case study) for embedded 'convective' or rapid ascent results in contributions to the total heating of 1.4%, 2.1%, and 4.9%, respectively. Thus, although rapidly

ascending WCB trajectories are heated significantly more than gradually ascending trajectories, at least in this case study, their total diabatic heating contribution is relatively small due to the small fraction of embedded convection. Relative contributions of embedded convective heating may, however, be more relevant in other case studies that are characterized by a larger amount of embedded convection than this WCB (e.g., Oertel et al., 2021).





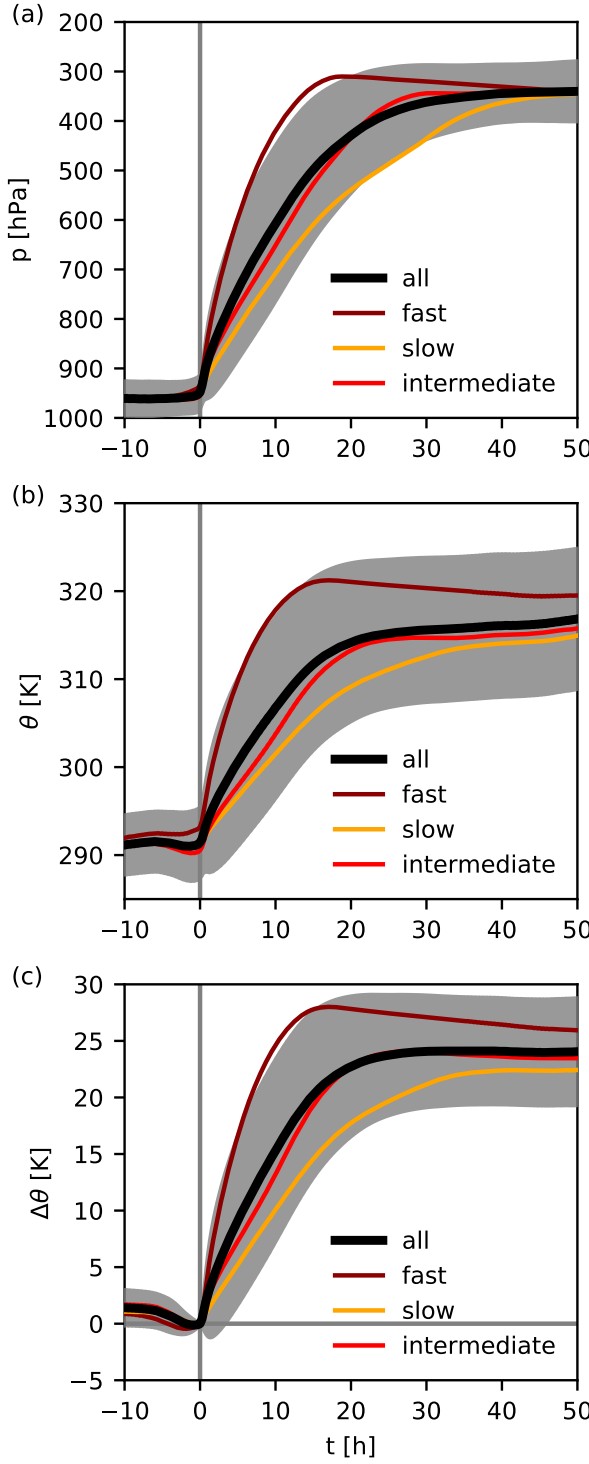

**Figure 6.** Mean evolution of **(a)** pressure (p, in hPa), **(b)** potential temperature ($\theta$, in K), **(c)** change of potential temperature since start of fastest 600 hPa ascent phase ($\Delta\theta$, in K) for all WCB trajectories (black line, grey shading shows $\pm$ standard deviation), and for the fastest 25% (dark red), intermediate (red), and the slowest 25% (orange).

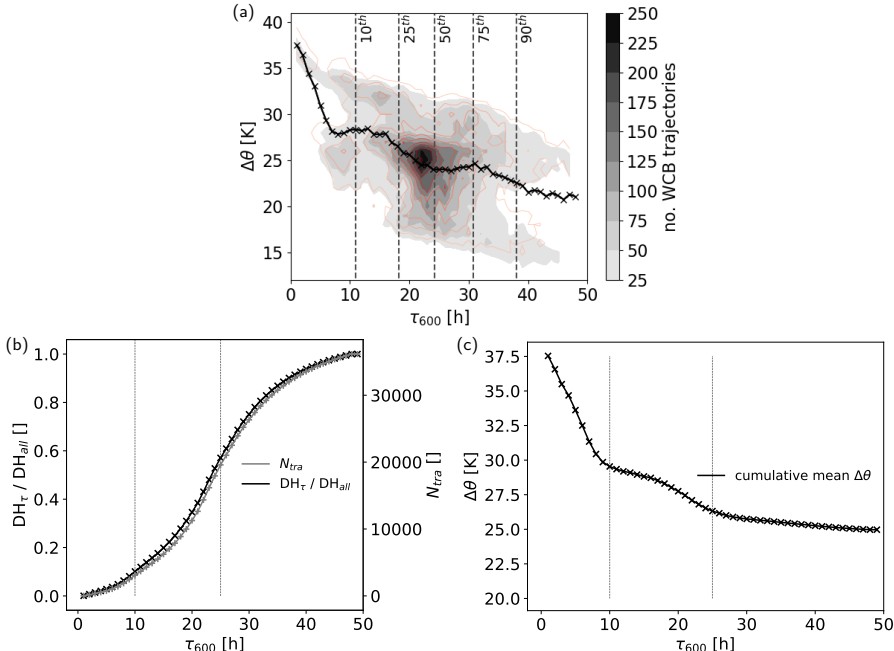

**Figure 7. (a)** 2D histogram of the duration of the fastest 600 hPa ascent phases ($\tau_{600}$, in h) embedded in all WCB trajectories and the change of potential temperature during this ascent phase ($\Delta\theta$, in K). Red contours show the maximum change of potential temperature along individual trajectories. The black line and markers show mean diabatic heating per $\tau_{600}$ bin. The lines show the 10th, 25th, 50th, 75th, and 90th percentiles of $\tau_{600}$. **(b+c)** Cumulative distributions of 600 hPa integrated diabatic heating as function of the time scale for WCB ascent ($\tau_{600}$, in h). **(b)** Relative contribution to total diabatic heating (black) and WCB trajectory number ($N_{tra}$, grey), and **(c)** cumulative mean diabatic heating (black). The grey vertical lines are placed at $\tau_{600} = 10$ h and $\tau_{600} = 25$ h.



## 4.1 *ADH* from bulk physics processes

During their ascent, trajectories are heated and also cooled diabatically from a large variety of non-conservative processes, which together result in the large total diabatic heating discussed above. Figure 8 shows the evolution of $ADH$ along WCB trajectories. The mean integrated diabatic heating for individual physics processes at the end of each trajectories' 600 hPa ascent phase for fast, intermediate, and slowly ascending WCB trajectories is shown in Figure 9. On average, the total 600 hPa integrated heating ($\Delta\theta$) amounts to 25 K, whereby the fast WCB trajectories are heated 3.8 K more than the intermediate WCB

trajectories, and 5.5 K more than the slowly ascending WCB trajectories (Fig. 9a).

The by far largest diabatic heating contribution results from microphysical processes (Figs. 9a). WCB trajectories are rapidly heated through cloud formation during their ascent, and after traversing through the mid-troposphere, the heating rate slows down (Fig. 8a-c). Turbulent diffusion, heating from the convection scheme (deep or shallow convection only, depending on the treatment of convection in the domain where the trajectories are located; see Section 2) and short-wave radiation also lead to a

net heating of the trajectories during the ascent, while long-wave radiative cooling is important in particular when trajectories approach the cloud top in the upper troposphere (Fig. 8a-c and 9a). Diabatic heating from non-orographic and orographic gravity wave drag has a minor contribution and is, hence, not explicitly considered in the following.

As microphysical heating has by far the largest contribution to the total diabatic heating with on average 95% of total heating, it substantially determines the cross-isentropic ascent of WCB trajectories. The larger total heating for fast WCB

ascent compared to slow WCB ascent is strongly related to larger diabatic heating through microphysical processes (Fig. 9a). Cloud formation sets in early after the start of trajectory ascent, leading to continuous increase in accumulated heating by microphysical processes along the ascent. In contrast, prior to the ascent, when trajectories are located in the lower troposphere, they are markedly cooled by microphysical processes (Fig. 8a-c). Individual processes that contribute to the total microphysical heating are discussed in detail in the next section (Section 4.2).

In the lower troposphere, diabatic heating by turbulent diffusion is the second most important process associated with a net heating. Integrated over the entire 600 hPa ascent, net turbulent heating amounts to 1.4 K on average, and is stronger for fast WCB ascent (Fig. 9a). The turbulent diffusion scheme is primarily active ahead of the approaching cold front in the warm sector between the surface and approximately 4 km height and forms a vertically oriented heating-cooling dipole (Fig. 10c, black solid and dashed contours). Gradient Richardson numbers are large in the WCB ascent region [1], and thus, a pronounced

heating-cooling dipole was not expected, although a recent study by Wimmer et al. (2022, see their Figs. S3 and S4) also shows such a heating-cooling dipole below 600 hPa height in the WCB ascent region. We assume that the turbulent diffusion scheme reacts to the pronounced vertical heating from the first saturation adjustment (Fig. 10a) which is called before the turbulence scheme and instantly modifies the local temperature profile.

The average heating from the convection scheme is comparatively small and amounts to only 1.7% of total $ADH$ (Figs. 9a

and 8a-c), primarily because deep convection is treated explicitly in the inner-most nest, where most trajectories ascend. As outlined above, this does not imply that embedded convection is absent or irrelevant, as convective WCB ascent can be

---

[1]Only in the lowest 1 kilometer the gradient Richardson number drops below 1.





explicitly represented in the high-resolution nest. On average, $ADH_{con}$ amounts to 0.5 K, and is smallest for the fast WCB trajectories (0.3 K for fast ascent, compared to 0.5 K for intermediate, and 0.6 K for slow ascent). $ADH_{con}$ is larger for the more slowly ascending WCB trajectories as they remain longer in the lower troposphere, where the primary heating source from the (shallow) convection scheme is located (Fig. 11c, black contour).

While shortwave radiation has a small heating contribution of on average 0.5 K during WCB ascent, long-wave radiation substantially cools the ascending air near cloud top in the upper troposphere, and hence, reduces the isentropic outflow level of WCB trajectories. This upper-level cooling contribution is non-negligible and amounts to on average -2 K integrated over the 600 hPa ascent. As the upper-level WCB cloud band can be relatively long-lived and the WCB trajectories remain in the cloud for several hours, cloud top cooling continues to reduce potential temperature of the WCB air parcels even after they have finished their main ascent.

### 4.2  $ADH$ from individual microphysical processes

In the following, we discuss the heating contributions from the individual microphysical processes. Net microphysical heating is dominated by condensation in the lower troposphere, which is realized by a first saturation adjustment, and leads to a rapid warming at the onset of the WCB ascent (Figs. 9b, 8d-f). Condensation is strongest in the warm-phase region of the WCB as reflected by the diabatic heating from the first saturation adjustment ahead of the cold front (Fig. 10a, black contour), however, is also relevant in the mixed-phase region up to approximately 6 km height. In the upper troposphere, vapor deposition on frozen hydrometeors takes over and dominates the subsequent cross-isentropic ascent (Fig. 10b, black contour; Fig. 8d-f) Together, condensation (on average 17 K) and vapor deposition on frozen hydrometeors (on average 13 K) dominate the microphysical heating budget (Figs. 9b, 8d-f), as expected from previous case studies (Joos and Wernli, 2012; Mazoyer et al., 2021). Vapor deposition is additionally split into its contributions from deposition on ice, snow, and graupel plus hail, respectively. In total, integrated heating from vapor deposition on ice $ADH_{QIDEP}$ is largest with an average heating of 7 K along the ascent. It also continues at higher altitudes and in lower temperature ranges where depositional growth of snow is small (Fig. 8d-f). $ADH_{QSDEP}$ amounts to on average 4 K, while vapor deposition on graupel $ADH_{QGDEP}$ has the smallest contribution of on average 1 K. The Eulerian perspective illustrates that the heating maximum of depositional growth of frozen hydrometeors is located at around 6 km height (Fig. 10b, black contours). Although heating dominates along the ascending WCB trajectories, the forward-sloping shape of the cloud band results in sublimation of sedimenting snow between 3-5 km height at the Eastern edge of the cloud band. Cooling from sublimation also occurs in a small region above the cold front.

The initial riming process prior to the growth of rimed particles by vapor deposition is associated with an averaged heating contribution of only 0.7 K (Fig. 9b), which is partially related to the small value of the specific latent heat of freezing, which is released during riming. Riming takes place in the mixed-phase region of the cloud between 2-6 km height (Fig. 11a).

Another major contribution included in the microphysical budget is related to the second saturation adjustment, which is associated with substantial cooling of an average 5 K during WCB ascent (Fig. 9b), and thus, reduces the heating from the first saturation adjustment and vapor deposition. The second saturation adjustment forms a vertically oriented cooling-heating-cooling tripole in the lowest 7 km (Fig. 10c, black contour), which leads to a non-monotonic $ADH_{satadII}$ evolution when





trajectories ascend through this region (Fig. 8e,f). The cooling from the second saturation adjustment results from an evaporation of cloud droplets in sub-saturated conditions as a response to established non-equilibrium conditions from processes that are called between the first and the second saturation adjustment (i.e., the turbulence scheme and all explicit microphysical processes; see Section 2.2). The cooling-heating dipole below 4 km height is to a large extent a response of the saturation
adjustment to the turbulence scheme, which is characterized by a dipole of opposite sign (cf. Fig. 10c,d). Hence, to a large extent, the diabatic heating and cooling from the second saturation adjustment in the warm phase can be understood as a quasi-compensation of the heating from the turbulence scheme (cf. discussion in Section 4.1). In contrast, the local cooling maximum of the second saturation in the mixed-phase cloud region above the 0°C-isotherm is to some extent the model representation of the Wegener-Bergeron-Findeisen process (Storelvmo and Tan, 2015), and reduces the substantial heating from
vapor deposition in the mixed-phase region (Fig. 10b,d).

    Evaporation of rain and melting of ice hydrometeors that are sedimenting into ascending WCB air represent important cooling processes. The total cooling through rain evaporation during the fastest 600 hPa ascent is relatively small and amounts to 0.4 K on average (Fig. 9b), and can arise during embedded phases of trajectory descent. Evaporative cooling is indeed important prior to the start of the WCB ascent, when trajectories are located in the inflow region in the lower troposphere below
the cloud band (Figs. 11b, 8d-f) where evaporative cooling dominates the total microphysical budget (Fig. 5d). Evaporative cooling within 10 hours prior to the ascent amounts to on average 2 K (Fig. 8d-f), and thus, can play an important role for the pre-conditioning of WCB air masses prior to their ascent. Melting from falling ice hydrometeors becomes important once trajectories approach the 0°C-isotherm, where the maximum cooling through melting is located (Figs.11c, black contour, 8b). Integrated along the WCB ascent, cooling from melting processes amounts to on average 1 K, and thus, has a comparable
magnitude to depositional growth of rimed particles. Overall, cooling from these processes offsets the strong heating from primarily condensational growth and can reduce the isentropic outflow level by on average 6 K along the ascent.

    The remaining microphysical processes explicitly represented in the two-moment scheme (CCN activation, freezing of cloud and rain droplets, homogeneous, deposition and immersion nucleation, as well as evaporation from melting ice hydrometeors) have comparably small heating contributions (Figs. 8g-i, 9c). Thus, in terms of total heating they are almost negligible, which
is primarily related to the small amount of mass conversion. Nevertheless, we hypothesize that the representation of these processes is important, because they influence the number concentrations of liquid and frozen hydrometeors (cf. Fig. B1), which subsequently influences the strength of other microphysical processes.

### 4.3   Relationship between WCB ascent behaviour and microphysical processes

Faster ascending trajectories are associated with larger $ADH$, which to a large extent results from stronger total microphysical
heating (Section 4.1, Figs. 7a, 9a). The latter is linked to on average larger initial moisture content of faster ascending WCB trajectories, which represents a thermodynamic constraint on the total diabatic heating from microphysics along the WCB airstream. A particularly large difference arises from the saturation adjustment. $ADH_{SATADI}$ for quickly ascending trajectories is 5 K larger than for slowly ascending trajectories. In contrast, $ADH_{QXDEP}$ differs less than 1 K between different ascent categories. Hence, in this case study the larger cross-isentropic ascent of the faster ascending WCB trajectories is substantially



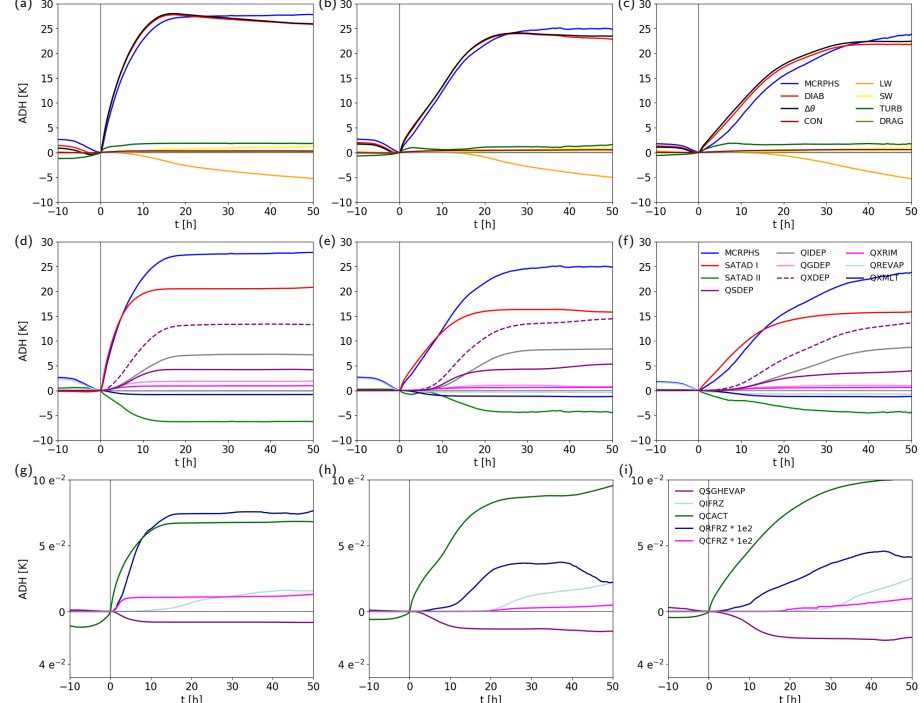

**Figure 8.** Mean evolution of accumulated diabatic heating (ADH, in K) along **(a,d,g)** fast, **(b,e,h)** intermediate and **(c,f,i)** slow WCB trajectories. **(a-c)** shows $ADH$ for turbulence (TURB), sub-grid scale orographic and graphity wave drag (DRAG), short- and longwave radiation (SW and LW), microphysics (MCRPHS), and the sum of all diabatic processes (DIAB), as well as the net change of potential temperature ($\Delta\theta$). **(d-i)** show $ADH$ from individual microphysical processes. **(d-f)** show first and second saturation adjustment (SATAD I and II), vapor deposition on snow, ice, graupel plus hail, and the respective sum (QSDEP, QIDEP, QGDEP, and QXDEP), riming (QXRIM), rain evaporation (QREVAP), and melting (QXMLT). **(g-i)** show evaporation from melting hydrometeors (QSGHEVAP), homogeneous and heterogeneous ice nucleation (QIFRZ), cloud droplet activation (QCACT), and freezing of rain (QRFRZ) and cloud droplets (QCFRZ). Note that QRFRZ and QCFRZ are scaled by 100. As in Figure 6, the x-axis shows hours since start of the fastest $600\,\text{hPa}$ ascent phase.

driven by larger initial moisture content of the rapidly ascending WCB trajectories, which is available for condensation. Considering the relative contributions of depositional growth of snow, ice, and graupel for the different WCB ascent categories additionally shows that for slower ascent vapor deposition on ice becomes increasingly more important. Prolonged residence time and enhanced hydrometeor growth in faster updrafts favours riming, thus, heating from particle riming and depositional growth of rimed particles is considerably larger for fast WCB ascent compared to slower WCB ascent. The representation

of graupel in the WCB is relevant as total diabatic heating from riming and growth of rimed particles on average exceeds $2.5\,\text{K}$ along fast ascending WCB trajectories. Besides the second saturation adjustment, melting of frozen hydrometeors and evaporation of rain have the largest microphysical cooling contributions during the ascent (Fig. 8d-f). Cooling contributions of evaporation and melting are both smaller for fast WCB ascent because WCB air parcels ascend faster through the relatively well-constraint vertical layers of the atmosphere where evaporation and melting are most pronounced (Fig. 11b,c, black con-





tours). In contrast, the slower ascending WCB trajectories remain longer near the surface and near the melting level, where rain evaporation and melting are strongest (Fig. 11b,c, black contours). Thus, the slowly ascending WCB air parcels are not only heated less, but also experience enhanced cooling from below-cloud evaporation and melting.

In the following, we consider processes associated with smaller mass conversion and related diabatic heating (Figs. 8g-i and 9c). Counter-intuitively, ascent-integrated heating from activation of cloud droplets (Fig. 9c) is larger for slow ascent, although

stronger updraft velocity is implicitly associated with larger supersaturation and an increase in CCN activation (eq. 1; Fig. 2b). Indeed, the initial increase of $ADH_{QCACT}$ is strongest along fast WCB trajectories, however, the prescribed activation rate of cloud droplets decreases quickly with altitude (Fig. 2b). The larger $ADH_{QCACT}$ for more slowly ascending WCB trajectories is attributed to the time integration as the slower WCB trajectories remain longer in the lower to mid-troposphere where CCN activation, as prescribed in the parameterisation, is most active (Fig. 2b). The latter is probably entirely dependent on the CCN

activation parameterisation, as no secondary CCN activation above cloud base would be expected at the typical evolution of updraft velocities along WCB ascent, which, however, is not accounted for in the ICON microphysics.

Freezing of cloud droplets and rain drops is more important during faster WCB ascent, as the timescale of transport in fast updrafts is smaller than the timescale of rain formation. Consequently more condensate is transported to colder temperatures and is available for freezing (Figs. 9c, 8g-i). In particular the homogeneous freezing of cloud droplets, which can produce large

ice number concentrations, influences the cloud structure via changes to the ice crystal number concentration. Homogeneous and heterogeneous freezing (immersion and deposition freezing) also have a negligible direct effect on the diabatic heating in the WCB due to the small mass conversion (Figs. 9c, 8g-i). Moreover, they mostly occur once WCB air parcels are finishing their main ascent. Evaporation from melting of frozen hydrometeors is associated with a small cooling of the ascending WCB air. Similarly to $ADH_{QCACT}$, the temporal integration accounting for the time trajectories spend in the region influenced by

evaporative cooling is decisive for total ascent-integrated cooling.

In summary, shorter ascent timescales (i.e., faster ascent) are associated with larger heating from the first saturation adjustment, riming, and freezing of rain, in contrast to vapor deposition which is reduced compared to longer ascent timescales (i.e., slower ascent). Simultaneously, long-wave radiative cooling, evaporative cooling and cooling from melting of frozen hydrometeors are reduced due to the shorter integration time scale. Vice versa, the slower WCB trajectories ascend, the less

important is heating from the first saturation adjustment and riming, and the more important becomes heating in the ice phase from vapor deposition compared to the mean over all trajectories. The larger diabatic heating from microphysical processes for faster ascending trajectories is related to enhanced specific humidity at the start of the ascent, which provides a thermodynamic constraint and emphasizes the relevance of WCB inflow moisture for total diabatic heating from microphysical processes and WCB trajectories' cross-isentropic ascent.

The heterogeneity of WCB ascent behaviour is in line with previous results highlighting the heterogeneous cloud structure of the WCB and its embedded convection (e.g., Crespo and Posselt, 2016; Oertel et al., 2019; Gehring et al., 2020). We further find that the heterogeneity is influenced by different microphysical processes that influence the cloud structure, but also latent heating and cooling along the ascent, which finally determines the total cross-isentropic ascent. Detailed representation of these processes in high-resolution NWP simulations allows to represent some of the variability in the cloud structure of WCBs.



The stratification according to ascent behaviour also emphasizes the relevance of WCB air parcels' residence time at a certain altitude for the total ascent integrated diabatic heating.



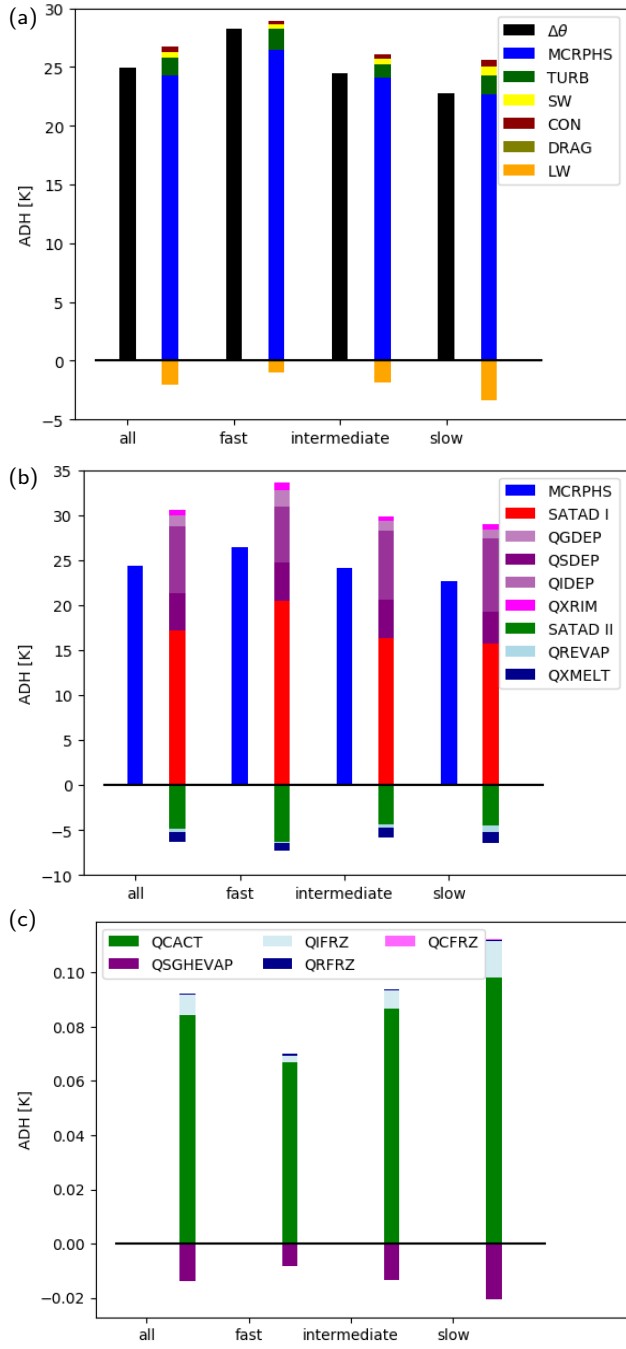

**Figure 9.** Mean $600\,\mathrm{hPa}$ ascent integrated diabatic heating ($ADH$, in K) for all WCB trajectories, and the fastest 25%, intermediate, and slowest 75% of all WCB trajectories for all processes shown in Fig. 8. **(a)** shows the net change of potential temperature ($\Delta\theta$) as well as $ADH$ for microphysics (MCRPHS), turbulence (TURB), sub-grid scale orographic and graphity wave drag (DRAG), and short- and longwave radiation (SW and LW). **(b)** shows $ADH$ for microphysics (MCRPHS) as well as first and second saturation adjustment (SATAD I and II), vapor deposition on snow (QSDEP), ice (QIDEP), graupel plus hail (QGDEP), riming (QXRIM), rain evaporation (QREVAP), and melting (QXMELT). **(c)** shows evaporation from melting hydrometeors (QSGHEVAP), homogeneous and heterogeneous ice nucleation (QIFRZ), cloud droplet activation (QCACT), and freezing of rain (QRFRZ) and cloud droplets (QCFRZ).



## 5 Diabatic PV modification

The strong ascent and diabatic heating of the WCB airstream typically results in low-level positive and upper-level negative PV anomalies (e.g., Madonna et al., 2014; Methven, 2015). Generally, the evolution of PV along the WCB trajectories (Fig. 12a-c) shows the well-known increase of PV during the first part of the ascent followed by a PV decrease later on (e.g. Wernli and Davies, 1997; Madonna et al., 2014; Joos and Wernli, 2012; Methven, 2015). On average, the WCB trajectories arrive in the upper troposphere near the tropopause with PV values below 0.5 PVU (Fig. 12a-c). The PV evolution along the ascent-centred trajectories shows a strong local maximum of 1-1.5 PVU immediately after the start of the ascent, which quickly decreases before reaching a plateau phase that is more pronounced for the intermediate and slowly ascending WCB trajectories and followed by further PV decrease. The initial PV increase along the ascent is much more rapid than shown in previous studies (e.g., Madonna et al., 2014; Joos and Wernli, 2012; Martínez-Alvarado et al., 2014), emphasizing the pronounced PV production in the lower-most 1 km (Fig. 11f) and its subsequent re-distribution. The strong initial PV increase is likely emphasized by centering the trajectories relative to the start of their fastest 600 hPa ascent and/or may be model and case dependent.

Integrating the PV tendencies from selected individual physics processes along the trajectories ($APV$) shows their contribution to the total change of PV (Fig. 12). The integrated effect of all diabatic PV tendencies ($APV_{DIAB}$, Fig. 12; neglecting PV change from momentum tendencies) approximately follows the evolution of total PV. The integrated effect of all microphysical processes only ($APV_{MCRPH}$) is responsible for the rapid PV increase in the lower troposphere, and also results in a subsequent decrease of PV. However, PV reduction is underestimated if the PV tendency from turbulent heat transfer is neglected (cf. $APV(\sum DIAB)$ and $APV(\sum MCRPH)$ in Fig. 12a-c and $APV_{TURB}$ in Fig. 12d-f).

Futhermore, differences between $APV_{DIAB}$ and the PV evolution partially result from neglecting the momentum tendencies (e.g. Spreitzer et al., 2019; Rivière et al., 2021), which can be important in the lower troposphere (Stoelinga, 1996; Attinger et al., 2021) and near the jet (Spreitzer et al., 2019). Further, the computation of PV tendencies on the original high-resolution ICON grid can result in very large tendencies as locally large diabatic heating gradients can occur in the simulation (see also Rivière et al., 2021). These artificially large tendencies are interpolated to the trajectory positions and can degrade the total PV budget (cf. Rivière et al., 2021; Wimmer et al., 2022).

Similarly to the diabatic heating contributions, the largest PV modifications along WCB ascent result from both saturation adjustments, vapor deposition, and turbulent diabatic heating (Fig. 12d-f). PV modification from rain evaporation substantially increases WCB trajectories' PV values, however, its effect is strongest prior to the start of the ascent near the surface (Fig. 12d-f), as evaporative cooling strongly increases PV in the lowest 1 km (Fig. 11b). Melting, riming, and radiative processes, as well as diabatic heating from the convection scheme contribute less to PV modification in the WCB.

$APV$ from the first saturation adjustment, vapor deposition and riming are characterized by the characteristic positive and subsequently negative PV rates (Fig. 12d-f), as the WCB trajectories pass through the respective local heating maxima and the related vertical dipoles of PV production and reduction (Figs. 10a,b and 11a). The increase of $APV_{SATADI}$ at cloud base is very sudden and pronounced. Its local maximum of PV production in the lower troposphere results from strong vertical



gradients of diabatic heating below 1 km height that occurs in an environment with large cyclonic vertical vorticity (Fig. 10a), which strengthens the diabatic PV production. Above 1 km height and above the local heating maximum, PV is reduced. The magnitude of PV reduction is smaller than PV production in the lower-most troposphere, but has a larger vertical extent.

Evaporation of rain strongly cools the boundary layer and lower troposphere below cloud base. The low-level cooling
results in PV production close to the surface, and at 1 km and 2 km height (Fig. 11b). Thus, $APV_{QREVAP}$ along trajectories increases considerably, with the largest increase of PV prior to the ascent when WCB trajectories are located in the lower-most troposphere (Fig. 8d-f). In total, PV increase from evaporative cooling along the WCB trajectories amounts to approximately 0.5-0.7 PVU. Hence, evaporative cooling has a substantial contribution to the PV maximum along WCB ascent and to the lower tropospheric positive PV anomaly.

PV modification from vapor deposition is important between 3-9 km height. The net PV change from vapor deposition along WCB trajectories is approximately 0 PVU, in contrast to the net positive PV change from the saturation adjustment, which depends on the ascent behaviour of the trajectories. Figure 10b shows the formation of a slightly tilted PV rate dipole, whereby PV reduction is stronger near the jet-facing side (eq. 3). Its formation results from horizontal heating gradients in a vertically sheared environment below the strong upper-level jet (Harvey et al., 2020; Oertel et al., 2020). Reduction of PV near the jet
region enhances the negative PV anomaly, and contributes to the PV gradient across the tropopause.

Similarly to their respective heating rates, the PV rates the second saturation adjustment and turbulence partially counteract each other, in particular below the 0°C isotherm. Due to locally large spatial heating gradients, the associated PV rates are very patchy (Fig. 10c,d). The vertically alternating patterns of strong PV reduction and production from the second saturation adjustment and turbulence (Fig. 10c,d) result from the vertical tripole of $ADH_{SATADII}$ and dipole of $ADH_{TURB}$, respectively
(Section 4.1). These PV rate pattern are reflected along the ascending WCB trajectories (Fig. 12d-f). Although both processes are associated with very large individual PV rates, their combined effect, and thus, their feedback to the dynamical core is relatively small, as these processes quasi-compensate each other. Hence, we recommend that detailed process studies using the ICON model consider the combined effect of $TURB$ and $SATADII$ if total microphysical heating budgets, including the second saturation adjustment, are considered.

The PV rates from the other microphysical processes are substantially smaller, as their associated spatial heating gradients are smaller. Similar to $APV_{QXDEP}$, $APV_{QXRIM}$ forms a vertical dipole of PV production and reduction, albeit with a smaller magnitude (Figs. 8d-f, 11a). As the diabatic heating maximum from riming is located near the 0°C isotherm, PV production and reduction takes place just below and above the melting level, respectively (Fig. 11a). In this region around the 0°C isotherm, the cooling maximum from melting of frozen hydrometeors is associated with PV reduction below 2 km height, and PV production
above (Fig. 11c). Accordingly, $APV_{QRMLT}$ decreases along WCB trajectories before it increases as trajectories pass through the melting minimum. The net PV change along the ascent is approximately 0 PVU, and the magnitude of $APV_{QRMLT}$ is larger than PV modification from riming.

The heating tendencies from the remaining non-microphysical parameterisation schemes are also associated with small PV rates. As diabatic heating from the convection scheme is small (Section 4), the associated $APV_{CON}$ is also comparatively
small (Fig. 12). The convective heating in the boundary layer forms a low-level vertical dipole of PV production and reduction





(Fig. 11e). Long-wave radiative cooling near cloud top and the associated PV rate dipole in the WCB ascent region are largest between 8-11 km height (Fig. 11d). The negative $PVR$ below the long-wave cooling maximum overlaps with the negative $PVR$ from vapor deposition, and together contribute to low PV values in the upper troposphere. While $ADH_{LW}$ continuously decreases once WCB trajectories approach cloud top, $APV_{RAD}$ depends on the WCB ascent behaviour and outflow height

relative to cloud top. Fast WCB trajectories also reach higher altitudes and traverse the cloud top cooling minimum, resulting in initial PV reduction below and PV production above the cooling maximum, respectively (Fig. 8d-f). The more slowly ascending WCB trajectories remain below the cloud top cooling maximum. Thus, $APV_{RAD}$ continuously decreases along their ascent.

In the cold sector of the cyclone (at relative longitude -3° to -6°), isolated shallow cumulus clouds form with cloud tops

between 1-3 km height (Fig. 11d). Radiative cooling by these shallow cumuli is one of the most important factors for PV change in the cold sector and contributes to the low PV values at low altitudes. Near the surface $PVR_{RAD}$ is negative in the cold sector and contributes to the negative near-surface PV values (Fig. 5b). $PVR$ from radiative cooling is partially compensated by opposing PV tendencies from the shallow convection scheme (Fig. 11e), which is active in the cold sector and forms a PV tendency dipole near the 0°C isotherm. These processes are not directly relevant for the considered WCB, however,

the pre-conditioning of the lower troposphere can influence potential successor cyclones and their WCBs (Papritz et al., 2021).

In summary, the different physics processes considerably modify PV during WCB ascent. While the evolution of PV along WCB trajectories is generally described as sequence of PV increase followed by PV decrease as trajectories pass through the heating maximum (e.g. Wernli and Davies, 1997; Madonna et al., 2014), our detailed analysis illustrates the complicated interplay and differing, partially opposing, contributions of all physics processes to the overall evolution of PV. The ascent-centred

Lagrangian perspective as well as the Eulerian composites emphasize the strong PV production in the lower-most troposphere by condensation and rain evaporation before PV values along the ascending WCB air parcels are reduced and eventually approximately approach their inflow values. PV reduction in the mid- to upper troposphere primarily results from condensation, vapor deposition, and longwave radiative cooling. Thereby, the PV evolution along the trajectories is influenced by the trajectories' ascent behaviour, i.e., the time the individual trajectories remain in regions which are predominantly influenced by

either PV production or reduction.

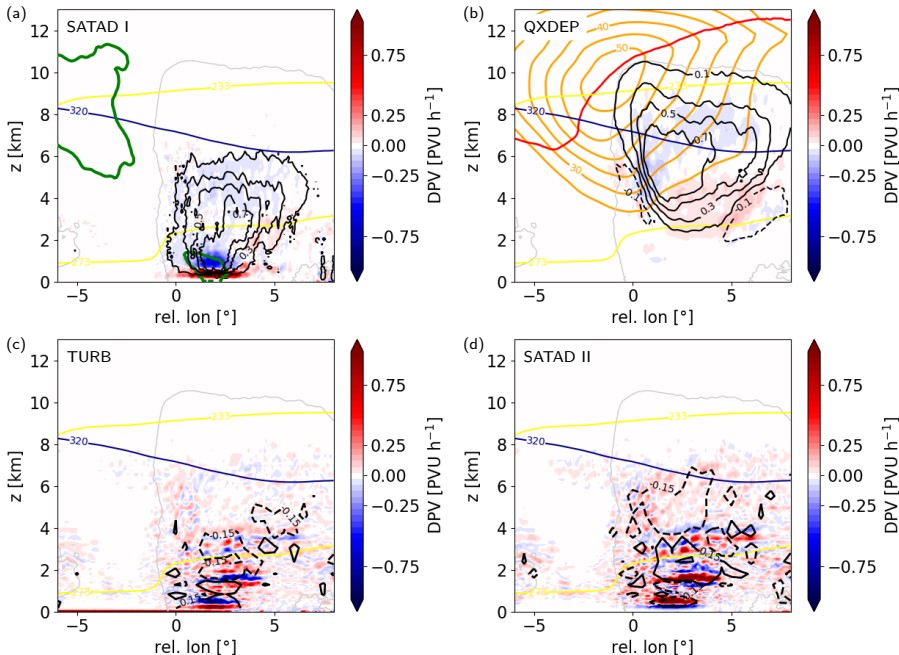

**Figure 10.** Vertical cross-section composites through the WCB ascent region ahead of the cold front for PV tendencies ($PVR$, in PVU h$^{-1}$ in colours) and diabatic heating (black, in K h$^{-1}$) from **(a)** first saturation adjustment (SATAD I), **(b)** total vapor deposition (QXDEP), **(c)** turbulent temperature tendency (TURB), and **(d)** second saturation adjustment (SATAD II). Also shown are **(a)** vertical vorticity (green line, only at $10^{-4}$ s$^{-1}$), and **(b)** wind speed (orange, every 5 m s). All panels show 273 and 233 K isotherms (yellow), 320 K isentrope (blue), and the cloudy region (grey line, at $q_x$=0.01 g k$^{-1}$). Note that the diabatic heating in **(c)** and **(d)** is interpolated to a coarser resolution for visualization of regions dominated by heating or cooling due the patchiness of the diabatic heating for these variables.

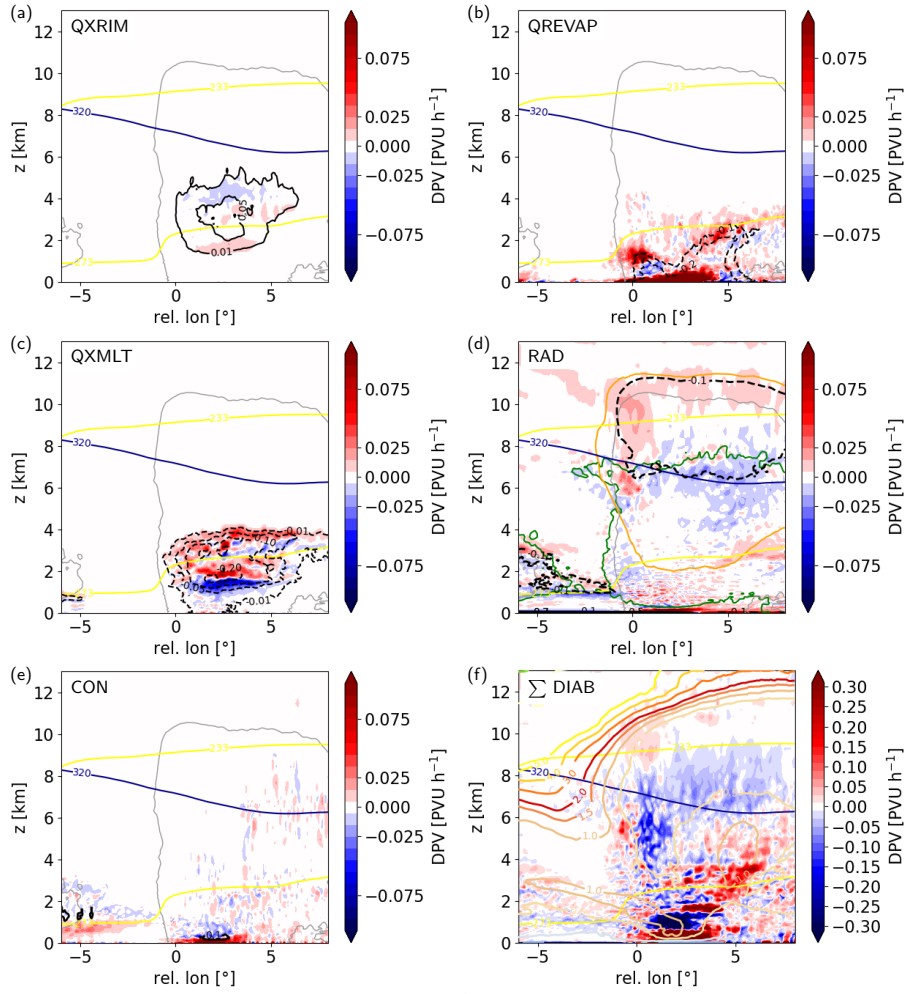

**Figure 11.** Vertical cross-section composites through the WCB ascent region ahead of the cold front for PV tendencies ($PVR$, in PVU h$^{-1}$, in colours) and diabatic heating (black, in K h$^{-1}$) from **(a)** riming, **(b)** evaporation of rain, **(c)** melting, **(d)** sum of long- and shortwave radiation, **(e)** convection scheme, and **(f)** sum of PV tendencies from all considered diabatic heating processes. **(d)** also outlines cloud and ice water content (green and orange contours, at 2 mg kg$^1$). All panels show 273 and 233 K isotherms (yellow), 320 K isentrope (blue), and the cloudy region ($q_x$=0.01 g k$^{-1}$, grey line). Note that PV tendencies in **(a-e)** are one order of magnitude lower than those in Fig. 10.

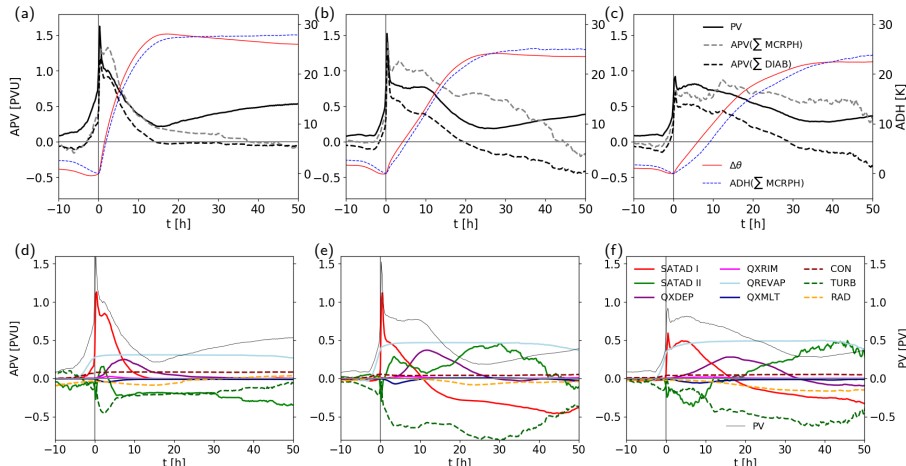

**Figure 12.** Mean evolution of PV and accumulated diabatic PV rates ($APV$, in PVU) along **(a,d)** fast, **(b,e)** intermediate and **(c,f)** slow WCB trajectories centred on the time of the fastest ascent. **(a-c)** PV as well as $APV$ for relevant microphysical processes ($\sum MCRPHS$; SATAD I and II, QXDEP, QXMLT, QXRIM, and QREVAP), as well as microphysics plus TURB, CON, and RAD ($\sum DIAB$). Accumulated diabatic heating ($ADH$, in K) from all microphysial processes (blue dashed) and the net change of potential temperature (red, $\Delta\theta$; i.e., total diabatic heating) are shown on the secondary axis. **(d-f)** $APV$ from individual diabatic processes: SATAD I and II, QXDEP, QXRIM, QREVAP, QXMLT, CON, TURB, and RAD. Full PV is shown on the secondary axis. As in Figure 6, the x-axis shows hours since start of the fastest 600 hPa ascent phase.



## 6    Concluding discussion

This study quantified the individual diabatic heating contributions that determine the cross-isentropic WCB ascent using online trajectories in a two-way nested ICON simulation of a case study in the North Atlantic, and additionally illustrated how the diabatic heating modifies the PV distribution in the ascending airstream. We provide a detailed ICON model perspective of microphysical process that are active in the WCB airstream and thus influence not only the WCB ascent but also the cloud and precipitation structure. To account for reduced aerosol loading in the North Atlantic region, the ICON simulations were performed with a modified CCN activation scheme. The implemented nesting support for online trajectories allows the flexibility to use higher resolution nesting without being too limited in terms of the size of the refined area, which is particularly important for larger-scale systems, such as WCBs, whose ascent covers a scale of 1000 km. In addition to quantifying the dominant (micro-)physical heating contributions for this WCB case study, we present two complementary perspectives, a Lagrangian and Eulerian perspective, that illustrate the interaction between detailed non-conservative processes and their effect on the PV distribution. The established Lagrangian perspective (e.g. Joos and Wernli, 2012; Joos and Forbes, 2016; Mazoyer et al., 2021) allows the quantification of processes that dominate the heat transport from the lower to the upper troposphere, and hence, determine the isentropic outflow level of the WCB airstream. The complementary Eulerian perspective gives an overview of where the most important diabatic heating takes place and how this modifies the PV distribution.

In agreement with previous case studies (Rasp et al., 2016; Oertel et al., 2019, 2021), we find a large heterogeneity in the WCB ascent behaviour, which not only affects the total ascent-integrated diabatic heating but also the heating contributions from individual (micro-)physics processes. The fast or convectively ascending WCB trajectories are heated substantially more than more gradually ascending trajectories, and thus, also reach a higher isentropic surface. However, their total heating contribution is still relatively small due to the small percentage of rapidly ascending WCB trajectories in this case study.

Consideration of all parameterised non-conservative processes shows that the ascent-integrated diabatic heating is dominated by microphysics. Diabatic heating from turbulent diffusion and the (shallow) convection scheme have a small contribution. Long-wave radiative cooling near cloud top has a non-negligible contribution, and decreases the trajectories' isentropic level after their ascent.

From all microphysical processes, the first saturation adjustment and vapor deposition on frozen hydrometeors dominate the total heating in the lower and upper troposphere, respectively, which is in agreement with previous case studies (Joos and Wernli, 2012; Martínez-Alvarado et al., 2014; Mazoyer et al., 2021). Riming only has a small diabatic heating contribution, due to the small latent heat of freezing. The ascending WCB air is cooled by melting of sedimenting frozen hydrometeors near the 0°C isotherm. Evaporation of rain in the lower troposphere substantially cools the WCB air, in particular prior to WCB trajectory ascent. The second saturation adjustment that is called after the explicit microphysical processes leads to a net cooling and is associated with the Wegener-Bergeron-Findeisen process in the mixed-phase region. We also show that the diabatic heating tendencies from turbulent diffusion and the second saturation adjustment partially compensate each other in the lower troposphere. Hence, we suggest that if diabatic heating from the second saturation adjustment is included in any microphysics budget, the partially compensating effects between turbulent diffusion and second saturation adjustment should



be additionally considered, in particular, because the individual heating tendencies of both processes locally can be very large, but tend to counteract each other. The remaining microphysical processes that are explicitly represented in ICON's two-moment scheme (such as ice nucleation and activation of cloud droplets) have negligible diabatic heating contributions, mainly because the mass conversion is relatively small. Although this has not yet been analysed systematically, we hypothesize that assumptions about the representation of these processes are still relevant, as they may subsequently influence processes with major heating

contributions, such as vapor deposition. For example, the modifications to the CCN activation scheme (Section 2.3), suggest that changes in CCN concentration also modify averaged heating rate profiles (Fig. B1d). Indeed, previous studies using coarser-resolution numerical models and including observations (Joos et al., 2017; McCoy et al., 2018) found small impacts of aerosol concentrations on WCB related precipitation and cloud structure. Future studies should more systematically analyse the influence of CCN activation on microphysical processes and cloud and precipitation formation in WCBs, similar to impact

studies of aerosol concentrations in convective environments (e.g., Seifert and Beheng, 2006b; van den Heever et al., 2011; Barthlott et al., 2017; Marinescu et al., 2021; Barthlott and Hoose, 2018; Barthlott et al., 2022).

While the detailed diabatic heating contributions, in particular from individual microphysical process rates, are model specific and vary depending on the parameterisation scheme (Joos and Forbes, 2016; Mazoyer et al., 2021; Wimmer et al., 2022), several studies using different WCB case studies, models and parameterisation schemes, have consistently suggested that

heating from condensation and vapor deposition are the dominant processes (Joos and Wernli, 2012; Joos and Forbes, 2016; Mazoyer et al., 2021). Therefore, we also expect that diabatic heating in other WCB case studies, which typically occur in an environment with strong dynamic forcing, is primarily determined by thermodynamic constraints. Thus, in future climate conditions the increase in specific humidity in the WCB inflow will results in enhanced diabatic heating and a higher outflow level (Joos et al., 2023).

We further analyse which PV tendencies strongly contribute to low-level PV production and upper-level PV reduction in the WCB, and thus, strengthen the prevailing anomalies. A local maximum of PV production is located in the lower-most troposphere, which is related to strong vertical gradients of the initial saturation adjustment, i.e. condensational growth of cloud droplets, in an environment with large vertical vorticity. Prior to the ascent, evaporation of rain near the surface also substantially contributes to PV production near the surface. These large positive PV values formed near the surface are

transported upward by the ascending WCB before the air parcels' PV values decrease towards their initially low PV values due to PV reduction from vapor deposition, longwave radiative cooling and condensation. Large integrated PV tendencies result also from the second saturation adjustment and heating from turbulence scheme. However, similar to their diabatic heating rates, the PV tendencies to some extent compensate each other. The additional microphysical processes such as evaporation, riming, and melting have smaller but non-negligible contributions to PV modification due to their smaller diabatic heating rates

and resulting smaller heating gradients. The relative contributions of the physics processes to the overall PV evolution depend on the detailed model representation of cloud and other physics processes and are therefore probably model-dependent.

The model representation of microphysical processes, in particular assumptions about aerosol concentrations or ice micro-physics, is associated with large uncertainties. It is yet open how sensitive WCB ascent and the associated cloud structure and precipitation formation are to the detailed representation of individual microphysical parameters, and how such uncertainties



propagate to the larger-scale. For example, it is yet open if environmental conditions, such as air pollution or dust outbreaks modifying CCN and IN concentrations, influence WCB ascent, or if the total ascent is insensitive to the detailed representation of cloud formation, as the bulk heating is dominated by the total conversion of specific humidity and therefore thermodynamic constraints. Therefore, sensitivity experiments employing the presented analysis framework will be conducted in future work, using varying assumptions about uncertain key parameters for WCB ascent. Such experiments can reveal if, and how, uncer-
tainties in the microphysics parameterisations influence WCB ascent behaviour, integrated-diabatic heating and propagate to the larger-scale flow. Additionally, extending the analysis to further case studies, and to different microphysics schemes and NWP models, would shed more light into the role of detailed representation of individual microphysical processes for WCB ascent, the cloud structure, and associated precipitation characteristics.

*Code and data availability.* The ICON source code is distributed under an institutional license issued by the German Weather Service
(DWD). For more information see https://code.mpimet.mpg.de/projects/iconpublic (DWD, 2015). The model output of the ICON simulation is available from the authors upon request. The reduced model output shown here will be permanently made accessible through the public KITOpenData repository https://bwdatadiss.kit.edu/) upon acceptance of this article. Information and the source code for the convolutional neural networks model ELIAS 2.0 are available from Quinting and Grams (2022). Size-binned aerosol number concentrations for the British FAAM flights during NAWDEX are available from the Centre for Environmental Data Analysis (CEDA) archive (CEDA, Centre
for Environmental Data Analysis, 2016).





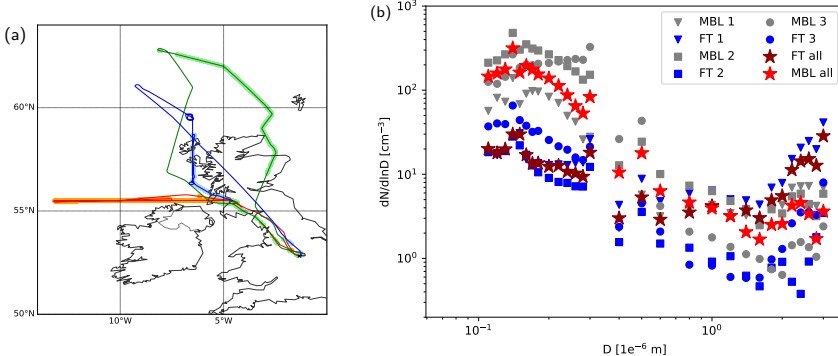

**Figure A1. (a)** Flight tracks for the NAWDEX British FAAM flights on 23 Sep 2016 (red), 27 Sep 2016 (green), and 14 Oct 2016 (blue). The highlighted tracks show regions where data is available. **(b)** Median log-normal aerosol size distribution for the marine boundary layer (MBL) and the free troposphere (FT) for measurements from all three flights and from the flights individually.

**Appendix A: Additional figure**





## Appendix B: Comparison of modified and original CCN activation scheme

The influence of the modification of the CCN activation scheme has been tested in a limited-area setup for the case study presented in detail above. Figure B1 shows averaged vertical profiles of hydrometeor number concentrations, water content, and selected diabatic heating rates in the WCB cloud band from two 24-h limited-area simulations using the (i) original and (ii) modified Hande scheme (in both simulations CCN activation was limited to temperatures above -38 °C). Both limited-area simulations were run for 24 h on the R03B09 grid and are initialized at 00 UTC 04 Oct 2016 from the ICON analysis. Lateral boundary conditions are provided every 3 h from ICON analysis fields. As expected, the CCN activation with the modified CCN activation scheme is reduced, in particular in the lower troposphere (Fig. B1c). This is associated with an expected reduction in cloud droplet number concentrations and cloud water content, while rain water content and number concentrations are increased (Fig. B1a,b). Although the effect on averaged vertical profiles of diabatic heating rates, such as condensation, evaporation, and vapor deposition on ice hydrometeors is relatively small (Fig. B1d), some difference in averaged diabatic heating rates are discernible that may influence WCB ascent (Fig. B1d).

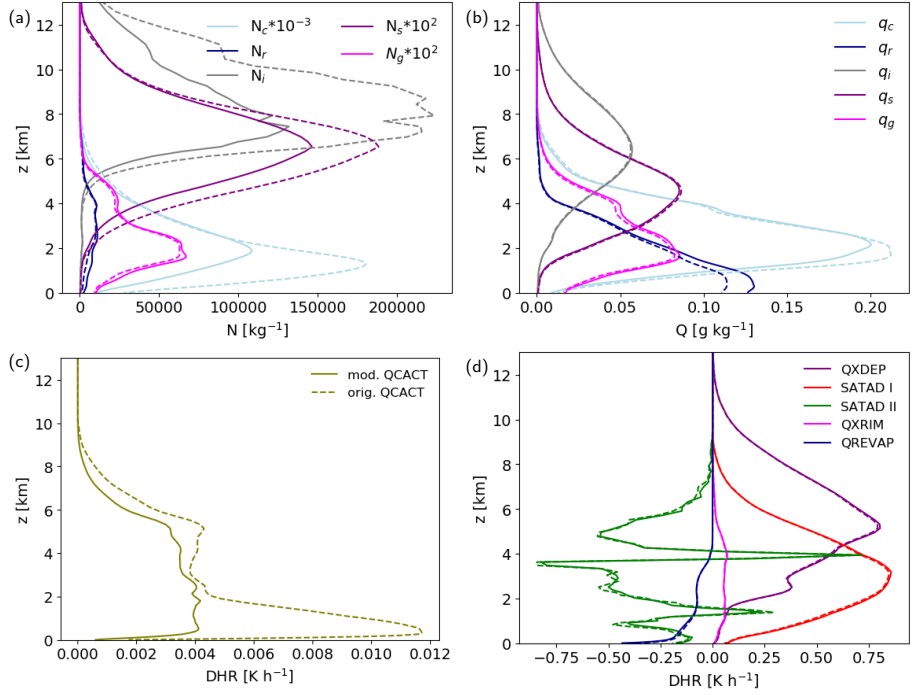

**Figure B1.** Averaged vertical profiles in the WCB cloud band (selected as region where total water path exceeds $1\,\mathrm{kg\,m^{-2}}$) from two 24-h long limited-area simulations of the WCB case study illustrating the differences that arise from a modification of the original CCN activation scheme. Solid lines show results from a simulation with the modified CCN activation scheme, and dashed lines show results from the original CCN activation scheme. Shown is **(a)** hydrometeor number concentrations (N, in $\mathrm{kg^{-1}}$)), **(b)** hydrometeor masses (q, in $\mathrm{g\,kg^{-1}}$), **(c)** diabatic heating rate from CCN activation (DHR, in $\mathrm{K\,h^{-1}}$), and **(d)** diabatic heating rates (DHR, in $\mathrm{K\,h^{-1}}$) from $1^{st}$ and $2^{nd}$ saturation adjustment (SATAD I and II), vapor deposition (QXDEP), riming (QXRIM), and evaporation of rain (QREVAP).



*Author contributions.* AKM and AO implemented the additional online diagnostics and performed the analysis. All authors continuously
discussed the results and contributed to the final manuscript.

*Competing interests.* The authors declare that they have no conflict of interest.

*Acknowledgements.* The research leading to these results has been done within the subproject B8 of the Transregional Collaborative Research
Center SFB / TRR 165 "Waves to Weather" (www.wavestoweather.de) funded by the German Research Foundation (DFG). CMG is supported
by the Helmholtz Association as part of the Young Investigator Group "Sub-seasonal Predictability: Understanding the Role of Diabatic
Outflow" (SPREADOUT, grant VH-NG-1243). The authors acknowledge support by the state of Baden-Württemberg through bwHPC.
The high-resolution ICON simulation was performed on the supercomputer Mogon-II at Johannes Gutenberg University Mainz, which is a
member of the AHRP (Alliance for High Performance Computing in Rhineland Palatinate) and the Gauss Alliance e.V. . The development
of the special diagnostics in ICON was additionally carried out on the supercomputers ForHLR II and HoreKa at Karlsruhe Institute of
Technology Karlsruhe, which are funded by the Ministry of Science, Research and the Arts Baden-Württemberg, Germany, and the German
Federal Ministry of Education and Research. The authors gratefully acknowledge the computing time granted on the supercomputers Mogon-
II at Johannes Gutenberg University Mainz and on ForHLR II and HoreKa at Karlsruhe Institute of Technology Karlsruhe. The authors kindly
acknowledge the CEDA data archive for providing access to the FAAM flight data.



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
