# Peer review of "Interaction of microphysics and dynamics in a warm conveyor belt simulated with the ICON model"

_EGUsphere, 2023_

## Referee Comment (RC2)

Review – Ambrogio Volonté

**Interaction of microphysics and dynamics in a warm conveyor belt simulated with the ICON model**

This manuscript contains a very extensive and thorough study that, using a case study, focuses on the effects of (total and individual) diabatic processes on the dynamics of warm conveyor belts. The manuscript is well-written and all the science questions are addressed in detail. My only issue is on the balance between the breadth of analysis included in the manuscript and the necessary synthesis of the results (see General comments). Therefore, I won't have any major problems with the publications of this work once the comments below are addressed.

**General Comments:**

I don't have any specific major comments on this work. My main concern is more general and relates to the risk of the main results of the analysis being somewhat hidden amongst the description of all the individual behaviours/effects/processes at play. I would suggest the authors go through the whole manuscript and see where and how the descriptions of the various figures can be slimmed down and the key results be better highlighted.

I think this is also true in the abstract and conclusions, both fairly long and detailed (too much?). I would like to see more emphasis on the motivation for this work, highlighting its novelty more clearly, and a more concise description of the results.

I must clarify that I see great value in being able to isolate the effects of individual diabatic processes in the thermodynamics and dynamics of WCBs and I really like the use of this combined Eulerian – Lagrangian framework. This is a great study. However, to avoid the risk of readers getting the impression that it is mainly "incremental" research, can you better highlight what is novel in its purpose, methodology and results? For example, could a final schematic help in highlighting that fast-ascending WCBs are also more diabatically heated and start with higher RH and relate this to implications for future WCBs? Or concisely point at the key modelling developments this study is suggesting?

The specific comments below are mostly very minor and/or typographic.

**Specific comments, line-by-line:**

Line 30: Most readers would probably know it already, but I suggest adding "warm-sector" before "cloud band".

Line 64: "in the dynamics of …"

Line 86: what does "The latter" refer to here?

Line 88: "major contribution to the PV modification" is this referred to both positive and negative changes or do we have different contributions to different parts of the heating pattern and subsequent PV dipole structure?

Lines 96-98: this is a fairly key point in providing motivation to this study. You could highlight it a bit more, and possibly include a shorter version of it in the abstract

Line 99: I suggest starting the sentence with: "Building upon previous research, in this study we provide…"

Line 100: "two-moment microphysics scheme": is this part of the novelty?

Line 126: Can you provide some background motivation as to why you are using a real case study (as opposed to idealised or free-running simulations) and why this case in particular? (I'm not at all against it and I know there are several good reasons, but would just like to see the argument made more explicitly) In particular, could you expand on why it is a good case even if, if I understand correctly, it's not a very convective one?

Lines 275-281: This is a very minor point, but I wouldn't use $\omega$ for relative vorticity in (3) and (4) when the very similar "w in italics" is used for vertical velocity at line 281.

Lines 309: What does "A" stand for in "ADH" and "APV"? If it's "accumulated", please say so.

Line 346: Use "60W", as the minus sign can easily be missed.

Line 365: "providing a high resolution in the WCB": word missing here? Also could you rephrase the following sentence? I think I got what you mean but it would definitely be clearer if you were to mention the two times and the associated in/out budgets separately.

Line 374: Do the composites include trajectories starting their ascent at different times? If so, did you consider focusing on a single time to reduce the smoothing effect of compositing? Please specify this in the text (linking it with the concluding sentence of the section)

Line 398: "considerably"

Lines 400-401: Any ideas on why there seems to be a "double path" for the slowest ascent, with either small (~15K) or large (25+ K) latent heating? (see Fig 7a) What is the setting leading to very slow but "very diabatic" ascent? How do you reconcile it with the statement at line 407 (and in the analysis following on)?

Line 621: "…the PV rates of the second…"

Lines 733-735: "yet open". It doesn't sound grammatically correct to me. I would replace it with "still not clear" or similar.

Fig 1:  Dark green crosses are not easy to spot

Fig 2b: The caption reads "CCN activation", while it's "CCN concentration" on the x-axis. I suggest making them consistent with each other.

Fig 3: It is rather difficult to see the green contours in panels a-c. Given that they are repeated in panels d-f, where they are far easier to see, I would consider removing them (or just keeping the outer one) from the earlier panels.

Fig 5: in caption (a), replace "isentropes" with "potential temperature" (or alternatively replace "temperature" with "isotherms", but I would prefer seeing the name of the physical variables rather than of the contours). In (c), could you add a small letter next to each of the species contours (e.g., 'r' for rain, 'c' for cloud, 's' for snow) or in a legend box, to help the readers visually making sense of it? Thanks!

Fig 10b: tropopause (red contour) missing from the caption.

Fig 11f: coloured PV contours missing from the caption.

---

## Author Comment (AC1)

**Response to reviewer comments for "Interaction of microphysics and dynamics in a warm conveyor belt simulated with the ICON model"**

Annika Oertel[1], Annette K Miltenberger[2], Christian M Grams[1], and Corinna Hoose[1]

[1]Institute of Meteorology and Climate Research, Karlsruhe Institute of Technology
[2]Institute for Atmospheric Physics, Johannes Gutenberg University Mainz

**Correspondence:** Annika Oertel (annika.oertel@kit.edu)

We would like to thank both reviewers for their overall very positive and constructive feedback. Following the suggestions of the reviewers the following major changes were made to the manuscript: We streamlined the manuscript and clarified the motivation for this study and for the modification of the CCN activation. To avoid distracting the reader from the main storyline the detailed methodology used to modify the CCN activation scheme was moved to the appendix. Furthermore, we included

5   a sketch summarizing the most relevant diabatic heating processes from Langrangian and Eulerian perspectives (Fig. 13 in the revised manuscript; Fig. 6 in this document) and improved the colours in the figures. Our detailed replies to all reviewer comments are included in blue below.

**1 Response to Gwendal Rivière**

**Comments by Gwendal Rivière** The paper is a numerical case study of a real extratropical cyclone. It investigates the role of individual microphysical processes in warm conveyor belt (WCB) heating and their impact on the circulation through modification of the potential vorticity. This problem has been only tackled by few studies before and needs to be deeper investigated in models with km-scale grid spacing and using state-of-the-art microphysical schemes like the two-moment scheme used in the paper. One original aspect of the study is to separate the heating and PV budgets between fast, intermediate and slow ascent trajectories. Such a decomposition is relevant since coarser resolution models do not represent fast ascent trajectories and it is important to check if such trajectories have an important effect on the whole heating budget. According to the present paper, the relative contribution of such trajectories in the heating budget is small because the number of such trajectories is small but is worth detailing the underlying processes because their proportion may vary from case to case.

The paper is well organized and very cleanly written. The choice and quality of the figures are very good. The cited literature is comprehensive. The heating budget along WCB trajectories is shown to be dominated by condensation of cloud droplets and vapor deposition on ice which confirm previous studies. The potential vorticity tendencies are positive in the lower troposphere due to condensation and vapor deposition but also due to melting and evaporation of rain. in the upper troposphere, PV is reduced via condensation, vapor deposition and long wave radiation. As previously said, the most original parts of the results come from the separated budgets of the fast, intermediate and slow ascents. An important effort is made by the authors to explain the CCN activation and the modification of the CCN activation scheme based on airborne measurements of aerosols. I am not expert on this latter aspect but it was not clear to me how the measurements were used to fit parameters of equation (1) determining the CCN number concentrations. My second concern is that the authors make an emphasis on this section fixing the right CCN number concentrations (section 2.3) but Fig.B1d clearly shows that this calibration of the scheme has almost no effect on the heating budget. So I am wondering if it is really necessary to put such a strong emphasis on this modification of the CCN activation scheme. Also in the conclusions (line 706), the authors mention CCN concentration modifies averaged heating rate profiles but it seems to be negligible looking at Fig.B1d. To conclude, I really enjoyed reading the paper, the results are well presented, some of them are original, some of them confirm previous studies, but the articulation of the CCN activation section with the rest of the paper is not clear to me. Therefore I recommend publication of the paper once the authors clarify the relevance of the CCN activation section within the whole paper and the conclusions related to the sensitivity of the heating and PV budgets to the CCN distribution. I have also added specific comments below.

We would like to thank Gwendal Rivière for his constructive and positive feedback and for his comments that helped to improve the quality of the manuscript. We reply to the general and specific comments point by point below.

**1.1 General comments**

1. The reason of choosing NAWDEX IOP7 is never said. Is it because some aerosols measurements are available from FAAM flights ? I am not expert on aerosols variability. But it is not clear to me if the calibration made of the CCN activation scheme duing weeks that are not overlapping with IOP7 case is relevant.

We chose this specific WCB case for several reasons. One of them was the availability of aerosol observations from the NAWDEX campaign, i.e., we selected a case from the North Atlantic during the campaign period (Sep/Oct 2016). The measurements did not take place exactly at the days of the WCB case. However, our intention of modifying the CCN activation was not to best represent daily variability in aerosol concentrations but to capture the general difference between continental and maritime aerosol concentrations (see l. 211 ff in the original manuscript). The CCN activation scheme in ICON's two-moment scheme is tuned for continental aerosol concentrations which differ substantially from North Atlantic conditions. As the observations show, there is still considerable (sub-)daily variability (Fig. A1b), which one can hardly capture with such a relatively simple CCN activation scheme and without prognostic aerosols. We tried to emphasize this motivation in the revised manuscript. To focus on key results, we moved the detailed description of the CCN activation parameterisation to the Appendix. Another reason for this specific case study was its association with the strongly amplified downstream ridge. A follow-up study, specifically addresses the impact of microphysical parameter perturbations on WCB ascent and the ridge downstream. We included a short statement on why we chose this case in Section 2.1: "The WCB case was selected as it is associated with a large-scale and strongly amplified downstream ridge (Section 3) and due to the availability of airborne observations of aerosol concentrations (Section 2.3)".

2. In the beginning of section 2.3, it would be good to present the objectives of that subsection and the different steps needed to tune the CCN activation scheme. It is difficult for a non specialist to understand the different steps. Since this section is quite long and the paper includes 3 figures on that aspect (Fig.2, A1, B1), the authors need to think which results listed in the abstract really depend on the modified CCN activation scheme. Would the results be the same with the original CCN activation scheme?

We believe that the results would be qualitatively similar, but details in, e.g., local diabatic heating, cloud structure, precipitation formation will differ (e.g., Fig. A1a,b). As mentioned above our main intention of modifying the CCN activation was to capture the general difference between continental and maritime aerosol concentrations. Our follow-up analysis explicitly addresses perturbations to the CCN activation scheme. We believe it is important that such perturbations are performed within a "realistic" parameter range; therefore, a baseline/reference simulation close to observations is important. Although the key take home message on diabatic heating contributions would not differ significantly without the adjustment to the CCN activation scheme, we think it is important that such adjustment must be documented. Moreover, it provides an alternative CCN activation scheme for the ICON model for maritime conditions and the opportunity to apply these adjustments in simulations which may be particularly interesting for the readership of ACP. We acknowledge that including this methodological part interrupts the main storyline, which is why we moved it to the Appendix. In the revised manuscript we now more strongly motivate the adjustments to the CCN activation and have clarified the respective necessary steps in more detail in the Appendix.

**1.2 Minor comments**

1. 1) Introduction: you may want to cite the recent paper of Mazoyer et al (2023, MWR) showing the sensitivity of the WCB heating rate and upper-tropospheric PV to the mixed-phase clouds representation using the one-moment scheme

75    of Meso-NH.

Yes, thank you. We added the recent study.

2. 2) Line 106: Which WCB case study ? So far, the case study was not introduced.

Thanks for spotting this detail. We rephrased the question to "a WCB case study".

3. 3) line 135: I do not see a separation between (i) and (ii). (ii) is included in (i) isn't it ?

80    It is correct that (ii) the ascent region is encompassed in (i). However, for (ii) the ascent region we include an additional higher resolution nest focusing only on this region. We re-phrased this statement, it now reads "The first refined domain encompasses the major part of WCB inflow, ascent and outflow region (R03B08 grid with approx. 6.5 km), and the second higher-resolution domain (R03B09 grid with approx. 3.3 km) is nested in the first one and focuses on the WCB ascent region (Fig. 1).".

4. 4) Figure 2: too small / dotted and dashed lines not really visible.

85    Thanks for this comment, we improved Figure 2 (larger font size, zoomed in, and increased line width). Please also note that Fig. 2a was moved to the Appendix.

5. 5) line 230: this is logical since the weather regime is zonal until late September and blocked between early October and late October.

90    Thanks for the additional information. We included it in the manuscript. "The backward trajectories show that the flights in September were influenced by long-range transport of maritime air advected from the North Atlantic during the transition to and presence of a zonal weather regime (Schäfler et al., 2018; Grams et al., 2017), while air measured during the October flight is potentially more strongly influenced by continental aerosol concentrations with low-level and mid-level air masses advected from the North Sea region during a Scandinavian blocking (Fig. A1a)." Please note

95    that this part has been moved to the appendix.

6. 6) Line 238: maybe one sentence to understand the purpose of using the cloud parcel model. to initilize which variable? How are used the observations? This might be straightforward for some readers but not for all of them.

In general, the parcel model simulates the activation of aerosols to cloud droplets given assumptions about aerosol size distributions and chemical composition as well as vertical velocity timeseries and initial thermodynamic conditions. In

100   contrast to ICON, a very short timestep is used that allows for an explicit representation of the supersaturation evolution, which is key to modelling CCN activation. The boxmodel simulations provide the final cloud droplet number concentrations (CDNC) at the end of the activation event (timescale typically a few to several 10 seconds) for each prescribed constant vertical velocity in the boxmodel simulations. From the observations, we take two fitted log-normal aerosol size distributions for the accumulation and the coarse mode (former Fig. A1b; Fig. A1c in the revised manuscript) and

105   assume their chemical composition. We included more detailed information in the Appendix.

7. 7) EQ (1) what are the constant parameters that are modified?

The parameters that are adjusted are $N_a$, B and C as function of pressure. $N_a$ is the number of aerosols as a function

of pressure and is estimated from the observations (assuming vertically constant aerosol number concentrations in the planetary boundary layer and an exponential decay with a scale height of 150 hPa (400 hPa) for the accumulation (coarse) mode in the free troposphere. The expression involving B and C describes the fraction of total aerosol being activated to cloud droplets at a particular vertical velocity. B(p) and C(p) are fitted to the CDNC data from the parcel simulations.

8. 8) Lines 260-263. According to Eq (1), CCN concentrations varies with w and p. So please be more precise when the word "varioability" is used.

    Thanks for this comment. Eq. 1 shows that the number of activated cloud droplets varies with p and w, i.e., with altitude and synoptic conditions. We meant to describe that the aerosol concentrations and size distributions can vary in addition to what is reflected in the still relatively simple parameterisation. To clarify, we replaced CCN concentrations by "aerosol concentrations".

9. 9) Figure 3, the yellow and green boxes are difficult to see

    Thanks, we changed the colour of the boxes and the colours of the background temperature field.

10. 10) Figure 5c, total is in shadings, isn't it ? where is the light blue contour ? the blue contour does not seem to be rain since it goes well above the 273K isotherm.

    This is correct, the total is in shading. We included this information in the caption. We apologize for the misunderstanding with the colours. The very dark blue is rain and the lighter blue is cloud water content. We changed the colours and included labels for the respective hydrometeors. Rain water content extends from the surface to slightly above the 273 K isotherm, which is (i) due to the compositing and varying heights of the 273 K isotherm with latitude, and (ii) due to rain not immediately freezing at 273 K.

11. 11) Is Figure 5 done with composites at different longitudes / latitudes but same time ? or different times ?

    Figure 5 is calculated using different longitude/latitude values and different times, including the main WCB ascent period (i.e., 04 / 05 Oct; see Section 2.6). This way we try to capture the spatio-temporal evolution of the WCB. For clarification we slightly re-phrased Section 2.6.

12. 12) Line 410: where is it shown?

    We do not explicitly show that "The faster trajectories ascend and also reach the upper troposphere further south ahead of the upper-level trough" in the manuscript. As the study focuses on the interaction between microphysics and dynamics we think that it is not necessary to explicitly show this in the manuscript. Figure 1 in this document shows WCB air parcel positions at the start of the fastest 600 hPa ascent (a) and when trajectories have finished their 600 hPa ascent (b). The colouring of WCB air parcel positions shows that the fast trajectories on average ascent further south and at earlier times in the cyclone life cycle (considering propagation of the cyclone to the north east). Moreover, these trajectories reach the upper troposphere further south due to their rapid ascent. However, not all trajectories that ascend in the south also ascend very rapidly. For clarification we added "on average" to not exclude slow ascent starting south and in the early phase of the cyclone.

[Figure]

[Figure]

**Figure 1.** WCB air parcel positions **(a)** at the start and **(b)** at the end of their fastest 600 hPa ascent coloured according to their ascent timescale $\tau_{600}$.

13. 13) It is not so clear that turbulence-related heating is larger for fast ascent. Maybe give the other values for comparison

    The differences are indeed relatively small, which we now mention in the revised manuscript (see l. 496 in the document with track changes). We also included the respective $ADH_{TURB}$ values.

14. 14) Line 498, why is there a cooling and not a heating ? We expect growth of ice crystals at the expense of liquid droplets in WBF process.

    Yes, this is correct. The mentioned local cooling maximum refers only to the diabatic heating rate (DHR) of the second saturation adjustment SATAD II (i.e., cloud droplet evaporation; Fig. 10d). In this region, growth of frozen hydrometeors (QXDEP) is associated with diabatic heating, which overcompensates the cooling from SATAD II (Fig. 10b). Thus, the net DHR is still positive (see Fig. 5d). We now more explicitly state that the local cooling is from SATAD II only, and not a net cooling.

15. 15) line 491 Does the evaporative cooling of cloud droplets only included in SATADII ?

    Evaporative cooling of cloud droplets can in principle also be included in SATAD I. However, in the WCB the general ascending motion does typically not lead to sub-saturated conditions in cloudy regions, i.e., there is no averaged cooling from SATAD I. ICON does not include any time-dependent parameterisation of cloud droplet evaporation (but only instantaneous evaporation via saturation adjustment in SATAD I and II), in contrast to rain evaporation, for which an explicit evaporation rate is implemented (QREVAP).

16. 16) Line 506: why is evaporative cooling term prior to ascent positive in Fig.8d-f ? I think this is due to the fact that accumulative diabatic heating is plotted from t=0 and that there is a decrease of accumulative heating from t=-10 to t=0 and in fact it represents cooling. Am I correct ?

[Figure]

**Figure 2.** Mean evolution of specific humidity along fast (dark red), intermediate (red), and slow (orange) WCB trajectories since start of fastest 600 hPa ascent phase.

Exactly, this is correct. To focus on WCB ascent only the diabatic heating is accumulated from t=0 h, i.e., cooling prior to t=0 h appears as heating. We added a clarifying statement at the end of Section 2.5.

17. 17) Line 508: why is Fig.8b here referred ?

Sorry, this is a mistake. We intended to reference Fig. 9b. This has been corrected, and for further clarification the figure reference was moved to the next sentence. Thanks. Please also note that we switched the order of Figures 8 and 9 in the revised manuscript.

18. 18) Line 525: is the moisture content of the different categories of trajectories before their ascent shown somewhere ?

No, sorry this has not explicitly been shown in the manuscript. The mean evolution of specific humidity is shown in Fig. 2. Figure 3 shows specific humidity at the start of the ascent ($t_0$) as function of the ascent timescale $\tau_{600}$ in 2 hourly bins. We included Fig. 2 as sub-panel in Fig. 6 in the revised manuscript and added a reference to the figure in the text.

19. 19) Line 529: please refer to figures. Fig. 9b?

We included the reference to Fig. 9b, thanks.

20. 20) Line 531: do you mean summing QGDEP and QXRIM ?

Yes, exactly. We included a clarification in the revised manuscript.

21. 21) Line 558: The total vapor deposition seems to me the same between the various categories of trajectories but the relative contribution of vapor deposition is smaller (Fig. 9b). I would change the sentence by replacing "vapor deposition" by "relative contribution of vapor deposition".

[Figure]

**Figure 3.** Mean specific humidity at WCB trajectory ascent start binned according to ascent timescale $\tau_{600}$ (2-h bins).

We included "relative contribution", although the absolute value for vapor deposition is indeed smaller for faster ascent. However, we agree that the relative contribution is the important difference here.

22. 22) I am surprised to see that the slight increase of PV after t=20h-30h (solid black curves in Fig.12a-c)is not captured by the sum of all diabatic heating processes (dashed black curves) which are negative at those times. Since the trajectories are in the upper troposphere at that time I do not expect PV tendency due to momentum non conservative processes to be important there. Do you have an idea of why such an increase in PV ? Could it be related to trajectories near the tropopause where PV gradients are strong and the Lagrangian trajectories are artificially crossing iso PV values ?

Both aspects probably play a role: The momentum tendencies from turbulence can be important also near the tropopause (e.g., Spreitzer et al., 2019). Moreover, it is plausible that the trajectories artificially cross iso PV values near the upper level jet where a strong PV gradient prevails. In the upper troposphere, where the vertical grid spacing is relatively large and where large PV gradients prevail, interpolation errors can arise and degrade the PV budget and also contribute to an increase of PV after the ascent. Nevertheless, we think that this slight increase of PV after the ascent is interesting and could be analysed in more detail in future work.

23. 23) Lines 609-614: there are some redundancies with lines 599-601.

Thanks for this detail! We reduced the information in the overview paragraph in l. 599-601 in the original manuscript.

24. 24) Line 616: do you mean that integration of QXDEP along WCB trajectories is near zero ? Looking at QXDEP in Figs.12d-f gives me the feeling that the integration should be positive. Even though the saturation adjustment leads to stronger positive peaks in DPV/Dt than QXDEP, the integrated values over the whole time of the saturation adjustment term seems to be small because after t=20h it is negative.

Yes, we indeed mean that integration of DPV/Dt of QXDEP along WCB trajectories is near zero. Fig. 12 in the manuscript shows the accumulated PV rates along the ascent, not the instantaneous rates. For example, for fast ascent (Fig. 12d) the average ascent-integrated PV change from QXDEP after the ascent is 0 PVU. Along the ascent PV is increased until t=9 h, and subsequently PV is decreased until it reaches 0 PVU. The saturation adjustment indeed leads to a net negative PV change after t=10-15 h for intermediate and slow ascent (for fast ascent it is also approximately 0 PVU). We removed the second part of the sentence referring again to the saturation adjustment because this is not relevant for the discussion of QXDEP.

25. 25) "PV rates FOR the second adjustment"

    Thanks for spotting this! We changed it to: "PV rates of the second adjustment".

26. 26) Lines 625-30 and also lines 495: I did not get why there is a counteracting effect of the second adjustment with respect to the turbulence ?

    We also find the interaction between TURB and SATAD II rather difficult, and believe that this interaction can be specific to the ICON model. Due to the interaction of TURB with the microphysics (cf. diabatic heating rates in Fig. 4a,b in this manuscript), we think it is relevant to also mention TURB in the manuscript. Our assumption about this interaction is the following: Through vertical mixing, TURB changes the temperature profile, and locally cools and heats mainly in the lower troposphere. This heating and cooling appears to be unbalanced with specific humidity. Thus, cooling results in supersaturated conditions; a heating, vice versa, to sub-saturated conditions. Thus, SATAD II evaporates cloud condensate (i.e., a cooling) in sub-saturated regions (previously heated by TURB) to establish thermodynamic balance. Vice versa for supersaturated conditions. The quasi-compensation of PV tendencies is analogous to the heating rates. We have not included too many details on this quasi-compensation as the underlying reasons for the model behaviour are not yet fully clear, however, included some more information in the revised manuscript.

27. 27) Surprisingly, the turbulence term is more negative than the radiative term in Fig.12 and is mainly responsible for reducing PV after t=10h (Fig.12 e-f). Comparing Fig. 10c (TURB) with 11d (RAD) does not provide the same picture as in Fig.12. Why ?

    The integrated PV tendencies from TURB as well as from SATAD II need to be considered very carefully. As the local tendencies of their DH can be very large, the PV tendencies locally can take very high absolute values, which strongly influences the mean of $APV_{TURB}$ and $APV_{SATADII}$. This accounts for averaging along trajectories as well as for the cross-section composites. In particular for the trajectories interpolation errors can arise from locally large gradients on the original grid (please see l. 687 ff in the document with track changes). Some of the very large tendencies arise from one single time step. As we are accumulating the tendencies online, one time step can already have a strong influence on the total $APV_{TURB}$ evolution. The strongest changes from TURB arise during the early ascent, i.e. before t=10 h (Fig. 12 d-f), which is consistent with Fig. 10c showing PV modification from TURB below approx. 5 km. Please not that Fig. 12 shows accumulated and not instantaneous PV rates. Concerning the contribution to negative PV from RAD: The change of PV along trajectories strongly depends on where the trajectories are located relative to the cooling maximum

230       as outlined in Fig. 11d. Only if trajectories continuously remain below the cooling maximum / cloud top, radiation can contribute to PV reduction. If they ascent to near cloud top (above cloud top cooling), they experience little PV reduction (or even PV production). Please also note that the colourbar scaling of PVR in Fig. 11 (RAD) is one order of magnitude smaller than in Fig. 10 (TURB). This was noted in the caption to Fig. 11. Fig. 4 in this document shows PVR of TURB as in the manuscript (Fig. 4c) as well as with the same colourbar scaling as PVR of RAD (Fig. 4d) and emphasizes that

235       despite the considerable averaging PVR of TURB is very patchy and can take locally high values, which unfortunately makes it prone to interpolation errors.

28. 28) Lines 660-665: I agree with most of the conclusions. But the sum of TURB and SATADII in Fig.12 seems to me to be the major source of PV reduction. This is in contrast with the cross sections of Fig.10 and 12 where we do not see any major negative PV tendency in TURB (Fig.10c). IN other words, I struggle interpreting the green dashed curve

240       associated with TURB in Figs.12d-f

      As mentioned in the reply above, the PV tendencies from TURB and SATAD II are influenced by single and locally very large values which influences the mean along the trajectories, and thus, makes it difficult to correctly interpret their mean evolution. Thus, in particular for SATAD II and TURB the mean PV evolution should be considered with care which is mentioned in the revised manuscript (l. 687 ff in the document with track changes). The composite perspective in Fig. 10

245       shows that in the lower troposphere large PV reduction occurs locally (e.g., at 1 and 2 km height). As trajectories ascend they experience alternating PV reduction and production by TURB at different heights. Although we average over a subset of coherently ascending trajectories this still results in slightly differing patterns along individual trajectories. In the summary paragraph, we did not explicitly mention the contribution from TURB, among others, as this study focuses primarily on microphysical processes. We think that the detailed contributions from TURB and its interaction

250       with SATAD II still needs to be better understood. This is, however, beyond the scope of this analysis.

29. 29) Since the original and modified versions CCN activation scheme lead to almost the same heating budget (Fig.B1d) and I imagine the same PV budget why is it so important in the paper to show the whole approach of CCN activation and its calibration with FAAM flights observations ?

      There are two reasons why we showed the adjustments in the current version of the manuscript: (i) Subsequent analyses

255       will focus on the sensitivity to systematically varying CCN concentrations of this case study (not only total heating budget, but also precipitation and cloud characteristics). For these experiments and the subsequent perturbations we want to use a more realistic reference and perturbations that are within the range of values reported in literature. The current version applied in ICON is valid for continental Germany, i.e. much higher aerosol concentrations are assumed. (ii) For these experiments, we think it is important that the applied method and results are properly documented. Nevertheless,

260       we decided to move the detailed description of the CCN activation to the Appendix (see also comments above).

30. 30) Line 687: according to Fig.12, turbulence is important for PV reduction (see above)

      Yes, we agree. TURB can play a local role for PV modification during WCB ascent. This sentence actually refers to the diabatic heating (DH). The relative heating contribution from TURB is small compared to all MCRPH.

[Figure]

**Figure 4.** Vertical cross-section composites through the WCB ascent region ahead of the cold front for **(a,b)** diabatic heating (DH, in $K h^{-1}$ in colours) for **(a)** second saturation adjustment (SATAD II) and **(b)** turbulent diffusion (TURB), and **(c)** PV tendencies (in $PVU h^{-1}$ in colours) and diabatic heating (black, in $K h^{-1}$) from the turbulent temperature tendency (TURB). **(d)** shows the same PV tendencies (in $PVU h^{-1}$ in colours) as in **(c)** but with a rescaled colourbar. **(b,d)** also show wind speed (green, every $5 m s$). All panels show 273 and 233 K isotherms (yellow), 320 K isentrope (blue), and the cloudy region (grey line, at $q_x$=0.01 g k$^{-1}$).

31. 31) Line 696: I would expect heating from WBF process ?! Why am I wrong ?

265    You are right, the net effect of the WBF, i.e., QXDEP plus SATAD II is heating (please see also reply to comment 14). We did not phrase this carefully enough. The net effect of SATAD II along the ascent is a cooling due to the WBF process, i.e., the net of the tripole of cooling-heating-cooling of SATAD II results in a cooling tendency. We rephrased this in the revised manuscript version.

32. 32) Line 706: I do not agree with this statement, the heating budgets are very similar between the two runs.

We agree that the difference between the averaged heating budgets is small. We rephrased this sentence and emphasize the differences in hydrometeor mass and number concentrations between both simulations.

33. 33) Line 707: Mazoyer et al (2021) used the same type of resolution as in the present paper and found a very small impact of aerosols (see their remark page 3965)

We included this reference, although Mazoyer et al., 2021 do not specify what "very little impact on the simulations" refers to, i.e., we are unsure if it refers to the entire evolution of the WCB, the cloud and precipitation structure, or individual heating rates.

34. 34) Line 718 "will result"

Thanks for spotting this typo.

35. 35) line 728: The compensation between second adjustment and turbulence is only partial in Fig.12 and the net effect seems to be a large PV reduction (same question as above)

Yes, this is correct. The net mean effect along ascent is a PV reduction (please not that we are not considering the momentum tendencies for PV rates). We explicitly state that they compensate each other only to some extent but also now include a sentence about their net negative contribution. As noted above, due to the locally large heating rate gradients, also the associated PV rates take extreme values which can introduce interpolation errors for the trajectories. A better understanding of the role of turbulence in the WCB requires further research which is however beyond the scope of this study. We slightly rephrased this part and include that the total effect is a PV reduction.

**2 Response to Ambrogio Volonté**

**Comments by Ambrogio Volonté** This manuscript contains a very extensive and thorough study that, using a case study, focuses on the effects of (total and individual) diabatic processes on the dynamics of warm conveyor belts. The manuscript is well-written and all the science questions are addressed in detail. My only issue is on the balance between the breadth of analysis included in the manuscript and the necessary synthesis of the results (see General comments). Therefore, I won't have any major problems with the publications of this work once the comments below are addressed.

**2.1 General comments**

I don't have any specific major comments on this work. My main concern is more general and relates to the risk of the main results of the analysis being somewhat hidden amongst the description of all the individual behaviours/effects/processes at play. I would suggest the authors go through the whole manuscript and see where and how the descriptions of the various figures can be slimmed down and the key results be better highlighted. I think this is also true in the abstract and conclusions, both fairly long and detailed (too much?). I would like to see more emphasis on the motivation for this work, highlighting its novelty more clearly, and a more concise description of the results. I must clarify that I see great value in being able to isolate the effects of

300 individual diabatic processes in the thermodynamics and dynamics of WCBs and I really like the use of this combined Eulerian – Lagrangian framework. This is a great study. However, to avoid the risk of readers getting the impression that it is mainly "incremental" research, can you better highlight what is novel in its purpose, methodology and results? For example, could a final schematic help in highlighting that fast-ascending WCBs are also more diabatically heated and start with higher RH and relate this to implications for future WCBs? Or concisely point at the key modelling developments this study is suggesting?

305 The specific comments below are mostly very minor and/or typographic.

We would like to thank Ambrogio Volonté for his positive feedback and the constructive comments to highlight the key results. Following the suggestion, we included a schematic of the most relevant microphysical processes in a WCB from Eulerian and Lagrangian perspectives (new Fig. 13 in the revised manuscript; Fig. 6 in this document) and more strongly emphasized our modelling developments in the methods (l. 332 in the document with track changes), and conclusions (l. 744), and abstract (l. 310 14 and 32 ff). Moreover, we streamlined the introduction and provided some more content on the relevance of this study for future sensitivity experiment and model development (l. 13-14, 111 ff, 740-742 in the document with track changes). Please also see reply to minor comment 5. The latter additionally motivates the study. To improve readability we also shortened figure captions where possible. We reply to the specific comments point by point below.

315 ## 2.2   Minor comments

1. Line 30: Most readers would probably know it already, but I suggest adding "warm-sector" before "cloud band".
   Thanks for the suggestion, we included "warm-sector".

2. Line 64: "in the dynamics of ..."
   Thanks, this is corrected.

320 3. Line 86: what does "The latter" refer to here?
   It refers to the smaller-scale PV dipole formation. We re-phrased this, it now reads "These smaller-scale PV anomalies are particularly relevant for".

4. Line 88: "major contribution to the PV modification" is this referred to both positive and negative changes or do we have different contributions to different parts of the heating pattern and subsequent PV dipole structure?
325   Both condensation and vapor deposition contribute to positive as well as negative PV tendencies along the WCB ascent - albeit at different heights.

5. Lines 96-98: this is a fairly key point in providing motivation to this study. You could highlight it a bit more, and possibly include a shorter version of it in the abstract.
   Thanks. We added some more information on how process understanding can contribute to targeted sensitivity experi-
330   ments. Moreover, we included a short statement in the abstract and conclusions.

6. Line 99: I suggest starting the sentence with: "Building upon previous research, in this study we provide..."

   *Thanks, we included your suggestion and additionally slightly rephrased this part. It now reads "Building upon afore-mentioned studies focusing on one-moment and/or quasi-two moment schemes, this study aims ..."*

7. Line 100: "two-moment microphysics scheme": is this part of the novelty?

   *Yes, partially. Previous studies have focused on one-moment or quasi-two moment schemes. To our knowledge, the individual microphysical heating and PV rates have not been investigated in the (ICON) two-moment scheme for a detailed WCB case study before. We added that previous studies focused primarily on one-moment and quasi-two moment schemes.*

8. Line 126: Can you provide some background motivation as to why you are using a real case study (as opposed to idealised or free-running simulations) and why this case in particular? (I'm not at all against it and I know there are several good reasons, but would just like to see the argument made more explicitly) In particular, could you expand on why it is a good case even if, if I understand correctly, it's not a very convective one?

   *The primary motivation for this study was to understand how microphysical processes in a real case of a WCB interact with each other and with the dynamics. Moreover, for generalization, follow-up studies should compare different WCB cases to provide a wider range of different cases. A comparison to an idealized simulation would, of course, also be very interesting. We specifically chose this cyclone and WCB as it was associated with the formation of a large-scale amplified ridge. In our follow-up study, we address the question of how parameter perturbations influence WCB ascent and the downstream flow evolution. Thus, the impact of the WCB on ridge amplification will be investigated in detail. Moreover, the availability of observations of aerosol number concentrations from the NAWDEX campaign motivated to consider a case study during the campaign period in the North Atlantic. We included a short statement on why we chose this case in Section 2.1: "The WCB case was selected as it is associated with a large-scale and strongly amplified downstream ridge (Section 3) and also due to the availability of airborne observations of aerosol concentrations". Finally, we are not primarily interested in a very convective WCB case. The presence of some embedded convection is advantageous but we did not intend to focus specifically on microphysics during convective ascent.*

9. Lines 275-281: This is a very minor point, but I wouldn't use $\omega$ for relative vorticity in (3) and (4) when the very similar "w in italics" is used for vertical velocity at line 281.

   *This is a good point, thanks for noting. We now use $\zeta$ for relative vorticity.*

10. Lines 309: What does "A" stand for in "ADH" and "APV"? If it's "accumulated", please say so.

    *You are right. It stands for "accumulated" diabatic heating and PV. We included it in the definition of ADH and APV.*

11. Line 346: Use "60W", as the minus sign can easily be missed.

    *We changed it to 60W, thanks.*

12. Line 365: "providing a high resolution in the WCB": word missing here? Also could you rephrase the following sentence? I think I got what you mean but it would definitely be clearer if you were to mention the two times and the

associated in/out budgets separately.

We slightly re-phrased both sentences. It now reads: "The ascent of the selected approximately 36 000 WCB trajectories predominantly takes place in the inner-most convection-permitting nest providing a high resolution for WCB trajectory computation (Figs. 3a-c and 4). Some WCB trajectories move into the highest resolution nest before or during their ascent (Fig. 4). Moreover, many trajectories leave the highest resolution nest once they reached the upper troposphere near the strong jet region.

13. Line 374: Do the composites include trajectories starting their ascent at different times? If so, did you consider focusing on a single time to reduce the smoothing effect of compositing? Please specify this in the text linking it with the concluding sentence of the section).

The composites include many time instances (04-05 Oct) of WCB ascent ahead of the cold front, i.e. it also includes several latitude/longitude positions moving with the cyclone's cold front and WCB. The time and positions are defined from the CNN-based WCB ascent mask (Section 2.6) which provides a smoother mask than re-gridding WCB trajectory positions and overall adequately represents the WCB ascent region (Fig. 3a-c). We also considered instantaneous vertical cross-sections and a shorter period for averaging during the analysis, however, this rather complicates the interpretation because at this relatively high resolution DH and PV tendencies are very patchy, small-scale and of high amplitude. We clarified the compositing method in this paragraph.

14. Line 398: "considerably"

Thanks for spotting this!

15. Lines 400-401: Any ideas on why there seems to be a "double path" for the slowest ascent, with either small ( 15K) or large (25+ K) latent heating? (see Fig 7a) What is the setting leading to very slow but "very diabatic" ascent? How do you reconcile it with the statement at line 407 (and in the analysis following on)?

The "double-path" in the figure is a bit misleading due to unfortunate choice of the colourbar thresholds. We now set the minimum number of trajectories per bin to 5 which shows that the space in between is also populated by trajectories and avoids the perception of a "double path". It is true that within each ascent timescale bin there is still substantial variability in $\Delta\theta$. Nevertheless, the general relation that faster ascending trajectories are on average heated more strongly still holds (see black markers in panel a and panel c). We included "on average" in line 407, and try to emphasize the variability more strongly in the revised manuscript.

The slow trajectories can nevertheless be separated in more strongly and less strongly heated clusters. To illustrate the difference between strongly and weakly heated trajectories, Figure 5 in this document shows the ascent start positions of slow trajectories with ascent timescales longer than 35 h and large (more than 22 K) vs. small (below 22 K) heating. The total heating is influenced by the location and time where trajectories ascend. The more strongly heated slow trajectories generally ascend further south and earlier than the less heated ones, i.e., trajectories that are heated less ascend at a later stage of the cyclone life cycle when the cyclone has propagated further north and climatologically less $q_v$ is available for subsequent heating (see also Figs. 2 and 3 and reply to comment 18 by Gwendal Rivière). Our following analyses are

[Figure]

**Figure 5.** WCB air parcel positions at the start their fastest 600 hPa ascent. Only slow trajectories are shown. More strongly heated trajectories are shown in red, less strongly heated trajectories are shown in blue.

based on the stratification into the three $\tau_{600}$ subsets and refer to the average of the respective subsets, although within the subsets there is still considerable variability. Despite this variability, the general conclusion that faster trajectories are heated more strongly than slower ones does not change.

16. Line 621: "...the PV rates of the second..."
    Thanks!

17. Lines 733-735: "yet open". It doesn't sound grammatically correct to me. I would replace it with "still not clear" or similar.
    Thanks for your suggestion, we re-phrased it.

18. Fig 1: Dark green crosses are not easy to spot
    Thanks, we changed the colours in Fig. 1.

19. Fig 2b: The caption reads "CCN activation", while it's "CCN concentration" on the x-axis. I suggest making them consistent with each other.
    The caption was adjusted and now reads "parameterised CCN concentration".

20. Fig 3: It is rather difficult to see the green contours in panels a-c. Given that they are repeated in panels d-f, where they are far easier to see, I would consider removing them (or just keeping the outer one) from the earlier panels.
    We decided to keep the "green" contours as this emphasizes the match between the trajectory-based and CCN-based approach to identify the WCB ascent region. For clarification we adjusted the colours.

21. Fig 5: in caption (a), replace "isentropes" with "potential temperature" (or alternatively replace "temperature" with "isotherms", but I would prefer seeing the name of the physical variables rather than of the contours). In (c), could you

[Figure]

**Figure 6.** Schematic **(a)** Eulerian and **(b)** Lagrangian perspectives of non-conservative processes in a WCB represented by the ICON model. Additionally, **(a)** indicates the Wegener-Bergeron-Findeisen process (WBF) which is represented through the interaction of vapor deposition (QXDEP) and the second saturation adjustment (SATAD II) as well as the quasi-compensation of SATAD II and the turbulent temperature tendencies (TURB) in the lower troposphere. **(b)** The colors along fast and slow ascent indicate the respective ascent-integrated diabatic heating anomalies and the font size indicates the diabatic heating strength of the individual microphysical processes. The 0°C isotherm is slightly elevated along fast WCB ascent.

add a small letter next to each of the species contours (e.g., 'r' for rain, 'c' for cloud, 's' for snow) or in a legend box, to help the readers visually making sense of it? Thanks!

Thanks for the suggestions. We changed isentropes to potential temperature and added labels to hydrometeor contours.

420    22. Fig 10b: tropopause (red contour) missing from the caption.

Thanks for spotting this! This is corrected.

23. Fig 11f: coloured PV contours missing from the caption.

Thanks again, this is corrected.

**References**

425    Spreitzer, E., Attinger, R., Boettcher, M., Forbes, R., Wernli, H., and Joos, H.: Modification of potential vorticity near the tropopause by nonconservative processes in the ECMWF model, J. Atmos. Sci., 76, 1709–1726, https://doi.org/10.1175/JAS-D-18-0295.1, 2019.